

**Mapping Rangeland Health Indicators in East Africa from 2000 to 2022**

Gerardo E. Soto[1,2], Steven Wilcox[,3], Patrick E. Clark[4], Francesco P. Fava[5], Nathan M. Jensen[6], Njoki Kahiu[6,7], Chuan Liao[8], Benjamin Porter[9], Ying Sun[2] & Christopher B. Barrett[10,11].

Affiliations:

[1] Instituto de Estadística, Facultad de Ciencias Económicas y Administrativas, Universidad Austral de Chile, Valdivia, Chile.

[2] School of Integrative Plant Science, Soil and Crop Sciences Section, Cornell University, Ithaca, NY, USA.

[3] Department of Applied Economics, Utah State University, Logan, UT, USA.

[4] USDA-Agricultural Research Service, Boise, ID, USA.

[5] Department of Environmental Science and Policy, Università Degli Studi Di Milano, Milano, Italy.

[6] The Global Academy of Agriculture and Food Systems, University of Edinburgh, Scotland.

[7] Department of Plant and Environmental Sciences, New Mexico State University, Las Cruces, NM, USA.

[8] Department of Global Development, Cornell University, Ithaca, NY, USA.

[9] Forest Ecosystem Monitoring Cooperative, University of Vermont, Burlington, VT, USA.

[10] Charles H. Dyson School of Applied Economics and Management, Cornell University, Ithaca, NY, USA.

[11] Jeb E. Brooks School of Public Policy, Cornell University, Ithaca, NY, USA

*Correspondence to*: Gerardo E. Soto (gerardo.soto@uach.cl)

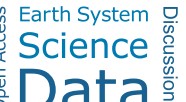

**Short summary**. Using machine learning for classification and linear unmixing, this paper produced Landsat-based time series of land cover classes and vegetation fractional cover of photosynthetic vegetation, non-photosynthetic vegetation, and bare ground in arid and semi-arid Kenya, Ethiopia, and Somalia. This dataset represents the first multi-decadal high-resolution dataset specifically designed for mapping and monitoring rangeland health
in the arid and semi-arid rangelands of this portion of Eastern Africa.



**Abstract.** Tracking environmental change is important to ensure efficient and sustainable natural resources management. Eastern Africa is dominated by arid and semi-arid rangeland systems, where extensive grazing of livestock represents the primary livelihood for most people. Despite several mapping efforts, Eastern Africa lacks accurate and reliable high-resolution maps of rangeland health necessary for many management, policy, and research purposes. Earth Observation data offer the opportunity to assess spatiotemporal dynamics in rangeland health conditions at much higher spatial and temporal coverage than conventional approaches that rely on in-situ methods, while complementing their accuracy. Using machine learning classification and linear unmixing, we produced Landsat-based time series from 2000 to 2022 at 30 m spatial resolution for mapping land cover classes (LCC) and vegetation fractional cover (VFC, including photosynthetic vegetation PV, non-photosynthetic vegetation NPV, and bare ground BG), two important data assets for deriving metrics of rangeland health in Eastern Africa. Due to scarcity of in-situ measurements in the large, remote, and highly heterogeneous landscape, an algorithm was developed to combine very high-resolution WorldView-2 and 3 satellite imagery at < 2 m resolutions with a limited set of ground observations to generate reference labels across the study region using visual photo-interpretation. The LCC algorithm yielded an overall accuracy of 0.856 when comparing predictions to our validation dataset comprised of a mixture of in-situ observations and visual photo-interpretation from very high-resolution imagery, with Kappa of 0.832; the VFC returned a $R^2 = 0.795$, $p < 2.2e-16$, and normalized root mean squared error (nRMSE) = 0.123 when comparing predicted bare-ground fractions to visual photo-interpreted very high-resolution imagery. Our products represent the first multi-decadal high-resolution dataset specifically designed for mapping and

monitoring rangelands health in Eastern Africa including Kenya, Ethiopia and Somalia,

covering a total area of 745,840 km$^2$. These data can be valuable to a wide range of

development, humanitarian, and ecological conservation efforts and are available at

https://doi.org/10.5281/zenodo.7106166 (Soto et al., 2023) and Google Earth Engine

(GEE; details in data availability section).

## 1. Introduction

Rangelands cover nearly half of the African continent land mass and support the livelihoods of

tens of millions of households (Reid et al., 2008, Sayre et al., 2013). The productivity of these

rangelands along with the human and livestock populations they sustain is significantly affected

by land degradation due to soil erosion, cropland expansion, shrub encroachment resulting from

heavy grazing and suppression of fires, as well as climate change and variability (Barbier and

Hochard 2018, Roques et al., 2001, Angassa and Oba, 2008, Wynants et al., 2019, Vetter 2005,

Hoffman and Vogel 2008). Episodes of extreme climate events, in particular, drought, have led to

emergency population migrations and humanitarian crises of historic proportions (Blackwell

2010). Improved understanding of the variation in rangeland health across space and over time is

crucial for community development, ecological conservation, and humanitarian programming in

the region.

The extensive development of Earth Observation (EO) platforms has largely improved our

understanding of ecosystems (Giuliani et al., 2020, Sudmanns et al., 2020). Long-term EO

systems, such as the Landsat constellation, have provided valuable data to assess and accurately

detect multiple ecosystem functions and patterns (Wulder et al., 2012, Loveland and Dwyer 2012,

Williams et al., 2006). Further development of EO and analytics has allowed the integration of multiple platforms into complex algorithms and workflows, benefiting from the ability of image data to scale at different spatial and temporal levels (e.g., AghaKouchak et al., 2015) and leading

to paradigm shift from change detection to continuous monitoring at high resolution (Woodcock et al., 2020). These recent developments have led to much interest in applying EO and related analytics to rangeland ecology and management (e.g., Allred et al., 2021, Hill et al., 2020, Rigge et al., 2020, Fava and Vrieling 2021).

Recent scientific advances create an opportunity to map rangelands health using satellite

imagery to monitor changes in rangeland health at ecologically meaningful scales for landscape planning and management (Allred et al., 2022). EO in these often-remote, arid and semi-arid regions becomes extremely valuable for its capacity to enable measurements in areas where data have never or rarely been collected on the ground. In addition, high-resolution (HR) remote sensing datasets can capture the fine spatial heterogeneity and the temporal dynamics that are key

to informing management decisions but are also exceedingly difficult to discern at scale using conventional, ground-based monitoring systems (Zhou et al., 2020).

EO-based data have been used to inform on rangeland health since the early days of EO programs (e.g., Landsat 1 program: Haas et al., 1975, Gaetz et al., 1976). Understanding of rangeland ecosystems relies on information about the specific composition of the various

vegetation communities within these ecosystems, oftentimes over large spatial extents, such as the Great Plains in North America (Reeves and Baggett 2014). Composition changes over time are important to track trajectories such as bush encroachment and soil degradation, impacts on grazers, etc. (Ghafari et al., 2018, Liao et al., 2018). HR thematic mapping of rangeland ecosystem can help explain key interannual variability in ecological processes such as water changes (Cooley et

al., 2017), terrestrial and aquatic vegetation phenology (Cheng et al., 2020, Coffer et al., 2020),

and crop dynamics (Lin et al., 2021), as well as long-term effects, such as land use change,

aboveground carbon, and sedimentation (Sankey et al., 2019, 2021).

The lower computational barriers from the continuous advancement of technology are

promoting the shift from plot-based assessments to the integration of satellite-based maps into

landscape management, improving broad-scale mapping of rangelands at higher spatial and

temporal resolutions than ever before (Jones et al., 2020, Allred et al., 2022). Many recent

contributions to this field have shown that even though moderate resolution datasets (from MODIS

sensors at 250 m resolution) are able to detect short-term vegetation phenology and long-term

demographic dynamics of herbaceous and woody species, they cannot detect changes at local

scales, because the spatial patterns of herbaceous and woody species typically occur at such fine

scales (Angassa, 2014, Browning et al., 2017, 2019, Matongera et al., 2021, Oba et al., 2003).

Despite collecting data at lower temporal resolutions, the Landsat collection at 30 m spatial

resolution has consistently played an important role in science for over fifty years due to

continuous efforts in calibration and corrections (Wulder et al., 2012, 2022, Franks et al., 2016).

The recent collection-based reprocessing that resulted in the Landsat collection 2 (Wulder et al.,

2022) represents an important opportunity to build consistent time series for HR rangeland

mapping. In addition, field studies have demonstrated that Landsat-scale sub-pixel estimation of

fractional cover of rangeland functional types, such as herbaceous and shrub components, and

especially bare ground, is crucial to overcome the difficulties of parsing out the underlying

heterogeneity within thematic land cover classifications and in understanding ecological dynamics

(Jones et al., 2018, Rigge et al., 2019). As a result, land cover classification (LCC) and vegetation

fractional cover (VFC, including photosynthetic vegetation PV, non-photosynthetic vegetation

NPV, and bare ground BG)) estimations have become the two building blocks of rangeland health

assessment of today's EO-based rangeland management (Jones et al., 2020). However, HR land

and fractional cover mapping (i.e., using Landsat) over large and remote regions is hampered by

the difficulty of collecting ground truth data at fine resolution. This is especially true in East Africa,

where limited infrastructure and physical insecurity make it very difficult to collect field data at

scale.

In this study, we produced a unique and new dataset composed of high resolution (HR)

LCC and VFC annual estimates of rangeland components for Eastern Africa based on the Landsat

collection from 2000 to 2022. We used a LCC scheme to help identify rangeland vegetation

transition pathways, and VFC to describe rangeland health condition trajectories within each class.

To overcome the challenge of scarce ground data for training and validating our models, for this

vast and remote region, we used a large collection of very high-resolution satellite imagery (VHR),

visual photo-interpretation and ad-hoc algorithms to generate a large sample of reference data to

generate and validate our two products.

## 2. Data and Methods

### 2.1 Study area

The study area is located in the semi-arid and arid regions centered on east and northern Kenya,

western Somalia and southern Ethiopia (Figure 1). We used two main features to bound our study

area. To the east and north, we used Landsat tiles, using PATH 164 and ROW 56 as limits,

dropping tiles PATH 164, ROW 59 and 60 due to heavy cloud cover. To the west and south, we

used a threshold value of mean annual precipitation of 700 mm using TerraClimate data (smoothed



with a kernel convolution with Standard Deviation = 5 km; Abatzoglou et al., 2018), thus keeping

the focus on the rangeland-dominated arid and semi-arid areas.

The study covers the epoch 2000 to 2022 to help capture decadal variation in ecosystem

conditions and maximize Landsat data availability. Landsat imagery is limited in this area due high

cloud cover often occurring during the two wet seasons observed in the region including the long

rains (March to June) and short rains (Oct-Dec). In most cases, cloud free data was available during

December through early March, which corresponds to the short dry (SD) season. Thus, we

generated our datasets using imagery from a portion of the SD, from 15 December to 1 March over

2000-2023, which maximized the annual available data count per pixel and ensured even

distribution of data over our period of study.


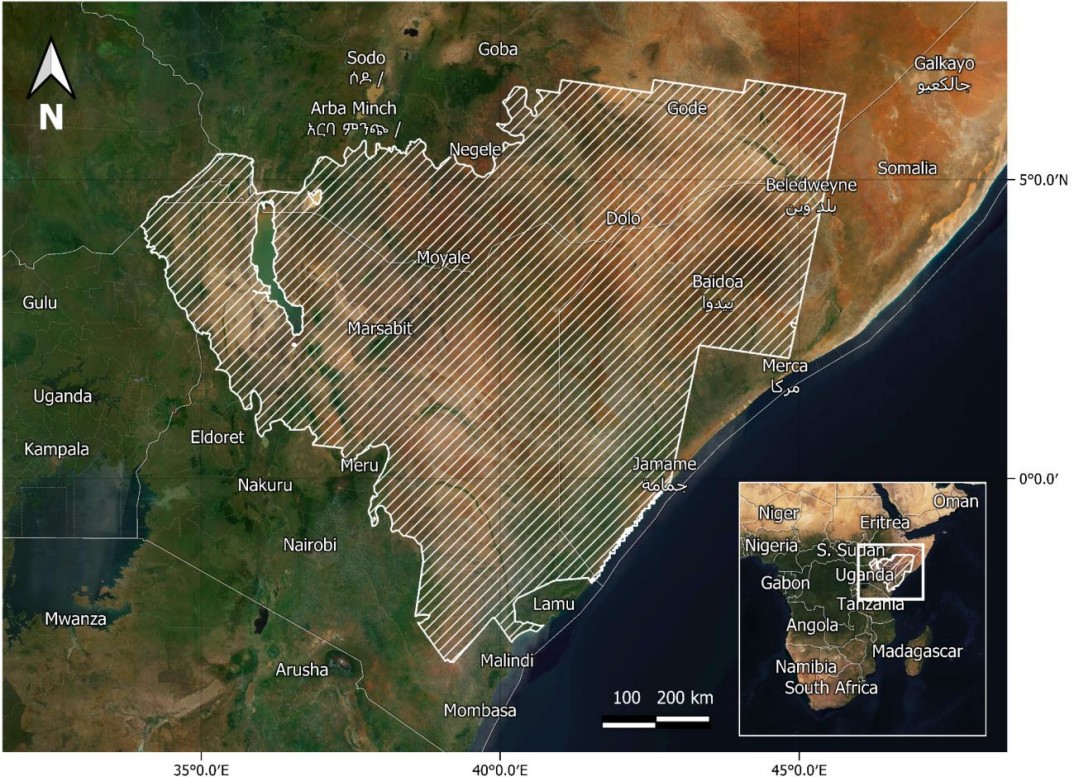

**Figure 1**: Map showing our study area in East Africa. Basemap: ©MapTiler, https://www.maptiler.com/copyright/.

*2.2 Remote sensing data*

2.2.1 VHR - Very high-resolution satellite imagery

To train our models and validate the results, we used VHR satellite imagery as little ground reference information exists in this vast and remote region. We obtained a large collection of imagery from Maxar Technologies via the NGA: ordered with the following filtering parameters:

sun elevation > 45°, off-nadir angle < 40°, and cloud cover < 50 %. The HR collection was composed of 2,500 mosaicked strips of imagery scenes from Worldview-2 and -3 sensors (Figure 2). These mosaicked strips, typically 16.4 km in width, were delivered as orthorectified- and radiometrically-corrected bundles of eight bands including Coastal (400-450 nm), Blue (450-510 nm), Green (510-580 nm), Yellow (585-625 nm), Red (630-690 nm), Red Edge (705-745 nm),

Near-InfraRed 1 (NIR1, 770-895 nm), and NIR2 (860-1040 nm) at a spatial resolution of 184 cm for WorldView-2 and 124 cm for WorldView-3, and a panchromatic band at a spatial resolution of 46 cm for WorldView-2 and 31 cm for WorldView-3. Shortwave Infrared (SWIR) imagery (1195 to 2365 nm) collected by Worldview-3 with a spatial resolution of ~3.7 m was also used in this study.

175        After subsetting to the SD season, we manually selected 321 strips maximizing the spatial coverage and minimizing cloud cover, as most images with scattered clouds projected oblique shadows often resulting in < 10 % of pixels being usable for further analysis. These data corresponded to imagery acquired from 2016 to 2020. We considered using Quick Bird imagery from previous years, but data availability for our AOI was minimal.

Earth System
Science
Data


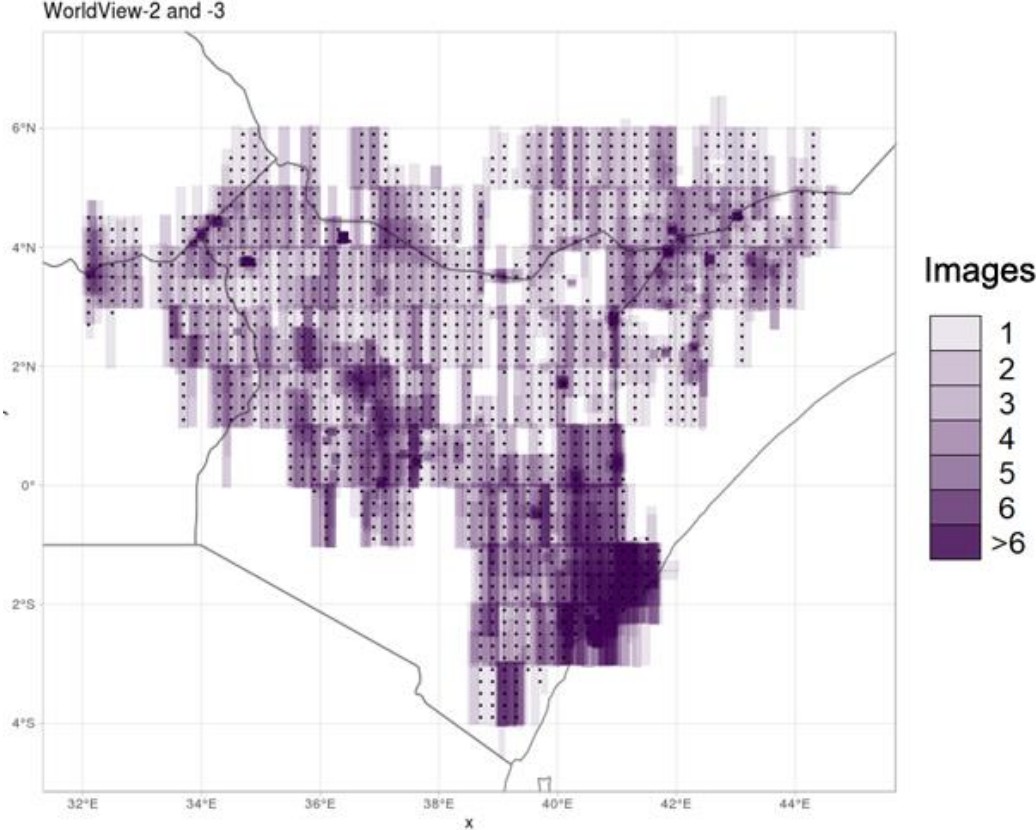

**Figure 2**: Spatial coverage of high-resolution imagery (polygons), and the spatial distribution of

point grid used for generating reference data.

2.2.2 HR - High resolution Landsat collections

To capture historical changes in vegetation health in our area of study, we utilized Landsat data

spanning over four decades ranging from Thematic Mapper (TM) and subsequent multispectral

sensors (1982-present; Wulder et al., 2012). While other studies have shown the value of higher

resolution sensors such as Sentinel-2 to show the potential higher gain in accuracy compared to

Landsat collection for the detection of invasive species in Eastern Africa (Duve et al., 2020), ESA's

Sentinel mission only features a short history of imagery acquisition from 2015 (Drusch et al.,

2013), which could bias our assessment towards the last decade, thus confusing the interpretation

of our results.

Landsat data is readily and freely accessible for scientific purposes. It available at different

processing levels, from raw images, to radiometrically-, geometrically-, and atmospherically-

corrected scenes (Wulder, 2019). We used Google Earth Engine (GEE; Gorelick et al., 2017) to

access and analyze atmospherically-corrected surface reflectance images for Landsat 5, 7 and 8

satellites from collection 2 (USGS, 2021), processed at the L1TP level

(https://www.usgs.gov/core-science-systems/nli/landsat/landsat-levels-processing). Landsat data

are packaged into overlapping "tiles", covering approximately 170 x 183 km each, using a

standardized reference grid (USGS, 2019). In this study we used 42 of these tiles, totaling

1,192,654 km$^2$. Differences in Landsat satellite sensors require different processing and correction

techniques. We describe each sensor first and then outline our harmonization efforts.

Landsat 8 Operational Land Imager/Thermal InfraRed Sensor (OLI/TIRS comprise of five

visible and near-infrared bands: Coastal aerosol, Blue, Green, Red and Infrared (NIR), two short-

wave infrared (SWIR1 and 2) and two thermal infrared bands (TIR). All bands were

atmospherically corrected using the LaSRC (Land Surface Reflectance Code; USGS 2020). Other

auxiliary data includes cloud, shadow, water, and snow masks layers generated with the C Function

of Mask (CFMask) algorithm version 3.3.1 and stored in the Pixel Quality Assessment Band

(QA_PIXEL; Foga et al., 2017, USGS 2022), as well as a saturation mask band in the Radiometric

Saturation Quality Assessment Band (QA_RADSAT).

Landsat 5 (TM) and 7 Enhanced Thematic Mapper Plus (ETM+) also contains two types

of observation bands according to their position in the electromagnetic spectrum. First, visible,

near-infrared and SWIR bands: Blue, Green, Red, Infrared (NIR), and SWIR1 and SWIR2 bands processed to convert raw values to orthorectified surface reflectance values. Second, one thermal

infrared (TIR) band processed to convert raw values to orthorectified brightness temperature values. All bands have a resolution of 30 m / pixel, with the TIR band collected with a resolution of 120 m / pixel (60 m / pixel for Landsat 7) but resampled using cubic convolution to 30 m. All bands were atmospherically corrected using LEDAPS (Schmidt et al., 2013). Other auxiliary data includes cloud, shadow, water, and snow masks layers generated with the CFMask algorithm and

stored in the QA_PIXEL band, as well as a saturation mask band in the QA_RADSAT band.

Landsat 7 has the potential to help fill the gaps between Landsat 5 and 8, being available from the year 1999 to date. However, the failure of the Scan Line Corrector (SLC) of Landsat 7 in 2003 somewhat limits its utility (Markham et al., 2004). This failure resulted in areas that are not imaged (~22 % of each tile), otherwise, data are valid for work and analysis. These data show

similar distribution of cloud cover and revisiting times as Landsat 8 collection. Hereafter, we refer to data pixels as any pixel where no masking occurred, and valid and usable data was available.

### 2.2.3 Landsat collection harmonization

We used reduced major axis regression to harmonize the surface reflectance values from Landsat 5 and 7 to match the spectral information of Landsat 8 following Roy et al., (2016) on each Landsat

data tile. These transformations are performed to improve temporal continuity between Landsat sensors (TM, ETM+ and OLI). After harmonization, the collections were merged and annual composites from December 15th to March 1st were generated using the median value of available data pixels. We used the median value, as the mean often gets biased with cloud contaminated pixels that was not included in the Level-1 QA_PIXEL Band used for cloud masking. In this study,

the year of the annual composites correspond to the calendar year where the composite starts (i.e.,

December 15th). We selected this time interval, as it was when imagery was mostly available, thus minimizing temporal imbalances among annual estimations. However, a large proportion of pixels were masked as a result of heavy cloud cover, with more than 5 % of masked pixels in 4 out of 21 different years. 2006 was a particularly problematic year, where the cloud component resulted in

10 % of pixels being masked (Figure 3).

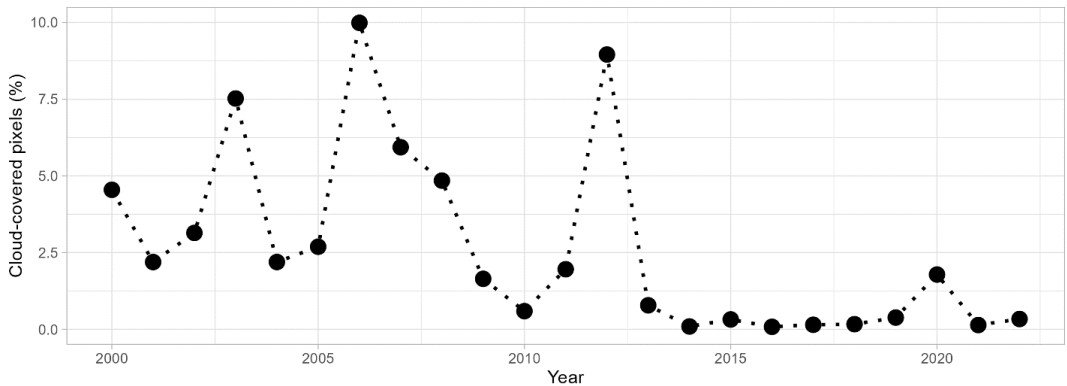

**Figure 3**: Cloud covered pixels present on the short dry season composites of Landsat imagery used in this study.


*2.1 Creation of reference data and general workflow*

The methodology to build long-term time series of LCC and VFC for rangelands in Eastern Africa is divided into three major steps. First, the development of a training/testing dataset from VHR imagery. Second, the LCC classification. Third, the VFC classification.

250         To integrate in-situ and VHR data to create reference data, specifically, we used ground reference data to inform a Visual Photo-Interpretation (VPI) protocol to create reference labels to train supervised classifications of VHR imagery. These VHR classifications were used to create a large amount of machine-generated reference data to train HR classifiers and to identify areas with



large proportions of the focal rangeland components for VFC estimation. To generate the LCC

reference data, we generated an algorithm that created reference points using a set of conditionals

with the proportions of reference compositional component (RCC), which includes vegetation

functional groups and other important classes such as bare ground, within each of our LC class

definitions, which were then compared to the calculated proportion of pixels from the VHR

classification within a moving window matching the 30 m spatial resolution of the HR data. We

also generated VFC reference data by using image segmentation on the RCC classifications with

the assistance of an application on GEE to identify homogeneous areas of rangeland components

that could spatially allocate HR pixels to use them to calculate spectral endmembers and generate

VFC estimations. Figure 4 shows our general workflow, including reference data partitions, remote

sensing data and results, processing algorithms, and accuracy assessments.


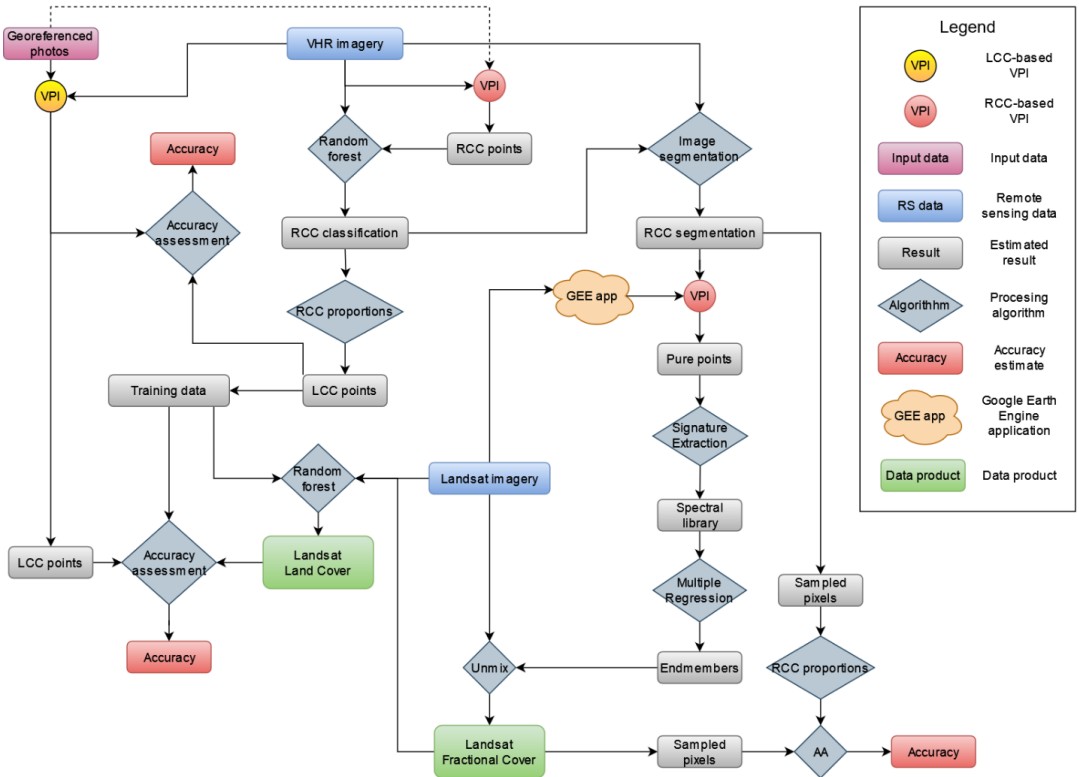

**Figure 4**: Schematic workflow of the process used in this work to generate Land Cover Classifications and Fractional Cover estimations based on Landsat imagery. The dashed line denotes the role of georeferenced pictures on informing the visual photo-interpretation (VPI) process. Short names correspond to: points generated with reference compositional component (RCC) work to train the RCC classification (RCC Points); collection of points with pure pixels (i.e., points with 100% of a single VFC type, Pure points); Library of spectral signatures of pure points (Spectral Library); Spectral endmembers for PV, NPV and BG (Endmembers); random sample of points with overlapping Landsat imagery to perform accuracy assessment on VFG estimations (Sampled pixels). See text for further details on the uses of data and processing.

*2.2 Development of training/testing datasets by integration of in-situ and VHR data*

2.2.1 VPI - Reference dataset by visual photo-interpretation of VHR imagery

We applied this classification scheme using VPI methods to develop training data for the classification algorithms for both LCC and VFC. We started first at the Borana Zone in southern Ethiopia, in the northern portion of our AOI where a rich source of georeferenced, ground-based photography (N = 1419 photos) was available for both a dry season 28 June – 26 August 2013 and wet season 6-31 May 2014 (Liao et al., 2018). In this VPI work, we leveraged this photography

with VHR satellite imagery of the same locations and approximate time frames to capitalize on the differing contextual strengths of each data source. The photography provided a low-angle oblique view of vegetation functional groups and canopy layers for better class identification. The VHR imagery, viewed via Google Earth (GE) or via United States National Geospatial-intelligence Agency's (NGA) Global Enhanced GEOINT Delivery (G-EGD), provided a broader, nadir-

oriented view of differing vegetation stands in context with one another, allowing more confident class separation.

        Specifically, a team of four VPI analysts followed a detailed protocol so as to ensure effective quality control. Training materials included reference flash card sets (see Appendix B) created for each of the eight land cover classes depicting a ground-based oblique view of a stand

of representative vegetation in addition to a nadir VHR satellite view of that same stand in context with other surrounding vegetation in the locale. Canopy cover flash card sets were also created for 2-m, 4-m, and 8-m shrub and tree crown diameters to aid in visually estimating cover percentages relative to the thresholds separating each land cover class. The VPI classification was calibrated using the reference card sets and a standardized set of VPI points and associated photographs and

imagery. Upon implementation, periodic spot checks of each analyst's VPI classifications were

conducted to affirm consistency and accuracy.

VPI classification took place as follows. The VPI point set from the georeferenced

photograph locations were randomly subset into equal partitions, and each partition was assigned

to a VPI analyst. The software package, Nikon View NXi ™ was used to view the photographs

and mapped camera location and oblique view direction on a satellite imagery background

provided by the software. The camera location coordinates were then plotted in GE and vegetation

at the location was evaluated using VHR imagery that was concurrent or nearly concurrent with

that of the photograph. Where concurrent imagery was missing from GE, imagery from the NGA

archive was ordered and viewed via G-EGD and a VPI-based classification was made for the

camera location. Where the camera location occurred in a mixed or ecotonal area, a new point in

a nearby, more representative location (i.e., more homogenous vegetation structure, cover, and

composition) was selected by the analyst and classified to a land cover class. Upon completion of

the VPI classification, a random sample of 10 % of the 1,419 VPI points was spot checked to

confirm overall consistency and accuracy across analysts. Where consistent bias or

misclassification was found, additional training was provided, and the analyst(s) re-visited all

assigned points for the troublesome class or classes and re-classified these points as necessary.

As the extent of this dataset was limited to the north area of our AOI, we extended the use

of this dataset as reference to inform recognition of the vegetation functional group components

of each land cover class used here. Vegetation functional groups generally refer to different types

of vegetation that are functionally and structurally different. In our setting, the primary groups are

trees, shrubs, and grasses. Using pan-sharpened VHR imagery, we then performed independent

VPI classifications of VFGs within classes to develop and refine a supervised machine classifier

and to support fractional cover analyses which are described in the next sections. This additional VPI work followed a procedure to spatially label the key components within each of the land cover

classes. These reference compositional components (RCC) included the vegetation functional groups (trees, shrubs, grass) as well as bare ground, water, cultivated land and impervious surfaces. We leveraged the combination of nadir views from VHR satellite imagery and the large set of available landscape photographs from the northern portions of our AOI to recognize visible characteristics of each sub-class component and apply these characteristics in VPI classification

of the entire study area.

### 2.2.2 RCC - Reference compositional component classification of VHR imagery

To create the reference dataset for calibration and validation of LCC and VFC estimations for our entire study area, we relied on RCC data generated from the classification of VHR imagery. RCC represents the basis of LCC as our land cover scheme (see below) follows a compositional

combination of them. In addition, RCCs are an important input for VFC estimation, which needs to be complemented with non-photosynthetic vegetation reference points, created with a different approach (see below).

We calculated the normalized difference vegetation index (NDVI) from the red and NIR-1 bands of the VHR imagery and then added the NDVI as a new band to the VHR dataset. Spectral

signals were then extracted and assigned to the points generated in the VHR VPI work with each assigned RCC class and a random forest classification was performed to predict RCCs using the spectral information as covariates. The number of trees was set to 1000, with two variables tried at each split. After model fitting, we used a graph showing the out of bag error of each class versus the number of trees in the classification to explore the effects of sample sizes on the accuracy of

the method and increase it when needed. Classification of VHR imagery focused on classifying

RCCs; trees, shrubs, grasses, bare ground, water, cultivated land and impervious surfaces (e.g., Figure 5). After training our classification algorithms on 90 % of the generated labels, we then used the remaining 10 % to compare the (out of sample, OOS) prediction of the classifier against the actual reference labels using confusion matrices. We set a threshold minimum value of 85 %

overall accuracy for using the resulting classifications in the following analysis steps. A random sample of VHR classified imagery with accuracies above the threshold was selected and visually inspected to understand misclassifications and their potential drivers. We increased our RCC-oriented VPI effort if threshold levels were not met, until accuracy met our threshold value. Despite our efforts and due to cloud cover and other factors such as cropland misclassifications in humid

areas, only 44.5 % (n = 143) of the total RCC classifications were retained using the 85 % accuracy threshold. Lower accuracy classifications occurred in areas of highlands on the west and southeast portions of our study area, characterized by higher precipitation. After contrasting classification predictions against pan-sharpened images, we recognized that most of the misclassifications corresponded to classes including green vegetation such as grass, crops and trees. Other sources

of error included areas with cloud shadows and impervious surfaces.

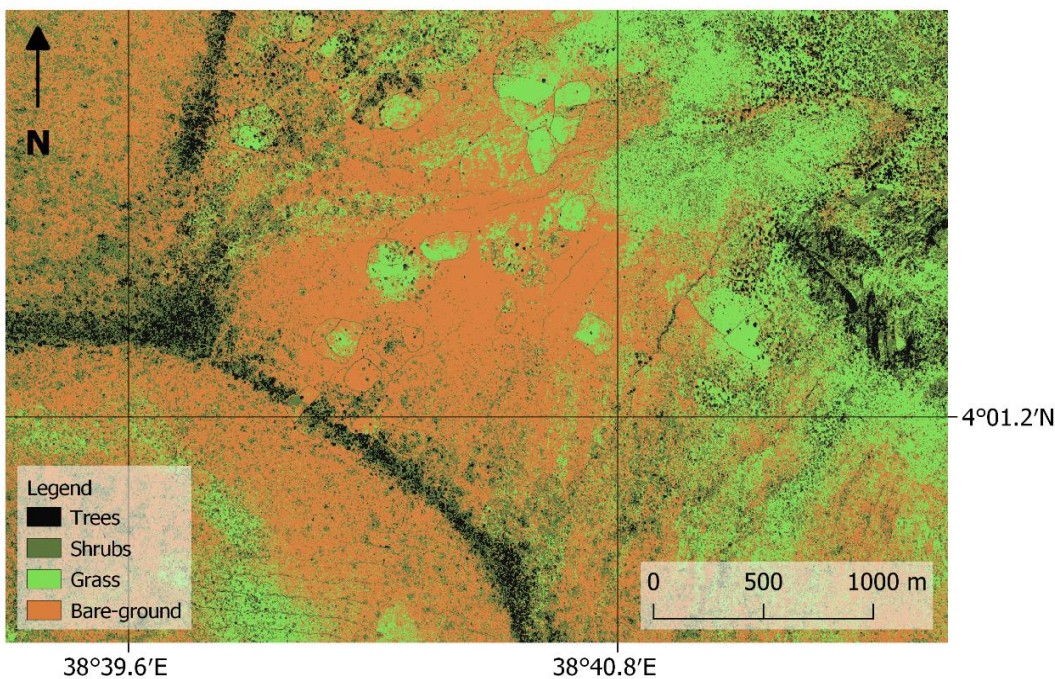

**Figure 5**: Example of a RCC classification result using a Worldview-3 image.

2.2.3 Composition-based algorithm for HR reference data creation

After classifying VHR strips and selecting those with higher accuracy, we applied a custom-made

algorithm that uses a squared moving window of the size of a Landsat pixel (30 x 30 m) and

calculates the proportion of VHR pixels, representing the area in the window covered by each of

the RCC classes from the predicted VHR classification. Using the proportion of VHR pixels for

each RCC class allowed us to use both Worldview datasets, as they have different spatial

resolution. Then, using the list of defined threshold compositional percentages of RCC classes per

land cover class in Table 1, we built code to meet the criteria for each land cover class. We then

selected a stratified random sample of 80,000 points to be used as training points for the Landsat

classification, described next. Points retained the date of the VHR strip used to generate them. Due

Earth System
Science
Data

to misclassifications associated with scattered cloud cover in some imagery, we further applied a buffer of 500 m around areas where more than 100 pixels of cloud or shadows were detected inside the moving window described above and excluded these from the RCC proportion calculation and class assignment.

*2.3 Land cover classification*

2.3.1 Land cover classification model

Our LCC scheme is based on the State Transition Model (STM; Bestelmeyer et al., 2017, Steele et al., 2012, Blanco et al., 2014) developed for this region by Liao and Clark (2018), with adjustments based on contributions from Pratt et al., (1966) and Liao et al., (2018) (Figure 6). Specific changes included the addition of classes not included in Liao and Clark (2018) and more

precise definitions of the characteristics of each class and the trajectories between them, given the extension of our study area. The scheme includes eight land cover classes, each representing a vegetation state defined by structure, cover, and functional group composition. The potential transitions among these states or classes are described in the mapping legend provided in Table 1, which adapts Table 1 from Liao and Clark (2018). However, tree, shrub, and herbaceous cover

thresholds have been further refined to better define class separations. The bushland class was also more clearly defined as a state where herbaceous presence was severely limited by climatic and/or edaphic factors rather than interspecific competition with shrubs and/or trees for resources. Transitional pathways associated with wild or prescribed fires have been excluded from Figure 6 and the legend (Table 1) to simplify description and presentation given the complexities associated

with fire-tolerant vs. fire-intolerant woody species, wildfire control, and past prohibitions on prescribed fire.

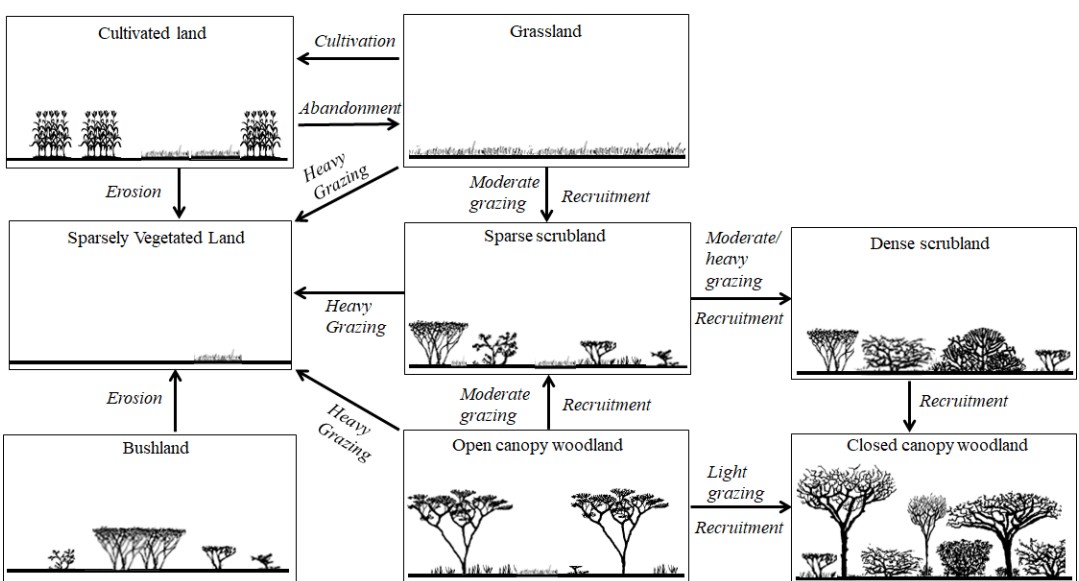

**Figure 6**: States and transition pathways among eight land cover classes.


**Table 1**: Land cover classes used for the Landsat land cover mapping (modified from Liao and Clark, 2018).

| Land Cover Class | Code | Description |
|---|---|---|
| Closed Canopy Woodland | CCW | Areas vegetated by a stand of trees with an interlaced canopy. Shrubs are usually present and interspersed within the woodland. Tree canopy cover is > 50 %. The CCW class represents a state that is usually distributed in relatively humid areas at upland elevations, on ridge crowns, or within riparian corridors where favorable edaphic and/or climatic conditions facilitate relatively dense tree growth. |
| Dense Scrubland | DS | Areas vegetated by an abundance of shrubs with a low to moderately productive herbaceous component. Shrub canopy cover is > 50 %. Herbaceous cover generally decreases with increasing shrub cover due to competitive relationships. Trees, if present are sparsely to moderately conspicuous with canopy cover typically < 10 %. The DS class represents a state to which the Sparse Scrubland (SS) state can transition to via shrub recruitment. The DS state can itself, transition |



| | | |
|---|---|---|
| | | to the Closed Canopy Woodland (CCW) state via tree recruitment under favorable edaphic and/or climatic conditions. |
| Bushland | BU | Areas sparsely to abundantly vegetated almost exclusively by shrubs. This class is largely limited to arid lowland areas where climatic and edaphic conditions severely limit herbaceous presence. Woody plant cover ranges 10-100 %. Although bushland thickets can and do form, this class represents a state that is separated from the Sparse Scrubland (SS) and Dense Scrubland (DS) states by its severe, site-based limitation on herbaceous presence. |
| Open Canopy Woodland | OCW | Areas vegetated by an open stand of trees with a sparse to abundant herbaceous or herbaceous/shrub component. Trees are always conspicuous occurring as scattered individuals or clumps of a few individuals with a canopy cover of 10-50 %. A woodland aspect is always retained. If shrubs are present, their occurrence ranges from sparse to common but shrub canopy cover is < 50 %. The OCW class represents a state which can transition to Sparse Scrubland (SS) by tree loss and shrub encroachment or, under favorable climatic and/or edaphic conditions, to Closed Canopy Woodland (CCW) by tree recruitment. |
| Sparse Scrubland | SS | Areas vegetated by scattered shrubs with a sparse to very abundant, productive herbaceous component. Shrubs are always conspicuous. Shrub canopy cover ranges from 10-50 %. Herbaceous cover generally decreases with increasing shrub cover due to competitive relationships. Trees, if present, are sparsely to moderately conspicuous with canopy cover of < 10 %. This vegetation class represents a state that lies between the Grassland (GR) and Dense Scrubland (DS) states. |
| Cultivated Land | CL | Areas currently being used for crop cultivation or, in cases of field abandonment, cropping disturbance is still visually evident (bare soil, tillage boundaries, etc.). Seasonally fallow fields (bare) are included in this class as well as those with growing crops. Common crops include *Zea s*p., *Sorghum sp.* and *Erogrostis tef.* CL areas usually fenced and located in places of relatively deep and moist soils (e.g., near seasonal river or stream courses). |
| Grassland | GR | Areas where the vegetation cover is dominated by grasses and occasionally other herbs. Herbaceous canopy cover ranges from 10 to 100 %. Widely scattered trees and shrubs may be present but woody |





| | | |
|---|---|---|
| | | canopy cover is < 10 %. The vegetation state represented by this class can transition to the Open Canopy Wood (OCW) state by tree recruitment or to the Sparse Scrubland (SS) state by shrub recruitment. |
| Sparsely Vegetated Land | SV | Areas poorly covered by vascular herbaceous or woody plants. Plant cover is < 10 %. SV typically represents areas where vegetation presence is severely limited by soil chemical (e.g., hypersalinity) or physical conditions (very shallow depth). Rock outcrops are included in this class. SV can also occur in areas which have suffered topsoil loss due to heavy disturbance (e.g., recursive, heavy grazing and/or trampling) and subsequent wind and/or water erosion. |

### 2.3.2 Land cover classification algorithm

The land cover classification consists of two general steps. First, the VHR imagery was classified

using the combination of RCC labels generated from VPI work (described in section 2.2.2) and

random forest classifiers (Belgiu and Dragut 2016), producing RCC classifications. Second, an

automatic algorithm, based on conditionals and the percentage thresholds of RCC defining each

LC class (described in Table 1) was run over the RCC classifications to generate new training

labels for the classification of the Landsat collections with a second random forest classifier. Here,

we describe the HR Landsat classifications.

Landsat collections were classified using the random points generated from the RCC

classifications (see section 2.4.1). We reserved 1,419 in-situ points from Liao and Clark (2018),

so we could later use this dataset with VHR ground reference data to independently assess the

accuracy of our results. We first masked all Landsat images using the SR_CLOUD_QA band

generated from the CFMASK algorithm of Surface Reflectance Landsat data. To eliminate water

bodies and rivers in our AOI, we applied a normalized difference water index (NDWI) mask,

whereby pixels with values > 0.2 were removed (Gao 1996). We also calculated and added

enhanced vegetation index (EVI), modified soil adjusted vegetation index 2 (MSAVI2), and

Normalized Difference Water Index (NDWI) bands to the collections (Qi et al., 1994, Liu and

Huete et al., 1995, McFeeters 1996). We also used CGIAR SRTM 90m Digital Elevation Database

version 4 to include elevation and derived slope and horizontal curvature (Jarvis et al., 2008,

Safanelli et al, 2020). Last, we included the bare ground and photosynthetic vegetation fractions

from our fractional cover results (see Figure 4) as covariates, which were found to increase

accuracy during our testing/tuning stage. We used the 80,000 algorithm-generated training points

through the RCC classification protocol explained in section 2.2, and randomly partitioned them

into 90 % training and 10 % for accuracy assessment. Then we extracted the spectral information

from the Landsat composite corresponding to the year of the date of each VHR image used to

generate the training points through VPI work (see section 2.4.1). With these points, we trained a

random forest algorithm to predict the vegetation classes of the entire collection. Thus, a single

multi-year random forest classifier was used for prediction on the harmonized Landsat collection.

After initial tuning of the classifier, we used 20 trees and a maximum number of 50 nodes. The

resulting classified collection includes images with pixel values associated with our main land

cover classes and masked pixels of cloud cover, shadows, and water.

2.3.3 Accuracy assessment of land cover classification

We used multiple reference year calibration to generate a classification model dependent on the

surface reflectance data (Gomez et al., 2016). Based on the standard assumption that surface

reflectance data represent the true ground response of features to sunlight, the classification model

is then used to predict past and future time steps in the RS time series. Often, these data are referred

to as absolute-normalized data (radiometrically and atmospherically corrected and orthorectified,

Thenkabail et al., 2015). After generating reference labels through the combination of VHR

imagery classification and an area-proportional classifier to upscale VFGs to land cover classes, we randomly partitioned this reference data set into training (90 %) and validation (10 %). We used the validation partition with the addition of the 1,419 points from Liao and Clark (2018) to

create confusion matrices to assess the accuracy of the predictions.

*2.4 Fractional cover classification*

We used bilinear unmixing to estimate fractional cover (Quintano et al., 2012) of three components of rangeland: bare ground (BG), photosynthetic vegetation (PV), and non-photosynthetic

vegetation (NPV). We combined VHR and Landsat imagery to identify homogeneous areas where the spatial footprint of Landsat pixels could capture pure spectral signals for the three components of fractional cover. In this context, pure refers to pixels with 100 % cover of one of our three main components (Boardman et al. 1995). Given the heterogeneity of soil types in our study area, we allocated special effort on finding as many BG pixels as possible. To find these, we used the

resulting RCC classification of VHR imagery (see 2.2.2) and performed image segmentation to identify homogeneous areas covered by bare ground. Because Landsat pixels are 30 by 30 m and their footprints could change with each revisit, we built an algorithm to scan the classifications to find homogeneous areas larger than 50 by 50 m, in order to allocate Landsat pixels with a margin of 10 m in both spatial axes.

460        We used GEE to manually create a sample of pure pixels, by mapping different Landsat color composites and creating graphs of 10-year-long NDVI and MSAVI2 time series and spectral profiles (i.e., spectral signatures) including all bands from the Landsat imagery for visually selected locations in the map. Using these visualizations, we checked that Landsat pixels corresponding to BG always covered the extent of the focal area and were not contaminated by

vegetation or other features such as litter or impervious surfaces. To identify PV, we checked

NDVI and MSAVI2 time series and natural color composites and selected a given acquisition time

for a Landsat image containing green vegetation. Finally, to identify NPV, we used the reflectance

profiles, NDVI and MSAVI2 time series and natural color composites to identify senescent

vegetation and pixels where and when crops were harvested and dead vegetation was left behind.

After a sample of 108 locations for bare ground, 900 locations for NPV, and 900 locations

for PV were established, the spectral information of the temporally closest Landsat image was

extracted for its use in the endmember estimation. We estimated the endmembers from the spectral

signatures of the sampled pure points using an R-based function for modeling of endmember

compositions based on bilinear unmixing (Seidel and Hlawitschka 2015, Weltje 1997). We used

"Blue", "Green", "Red", "NIR", "SWIR1", "SWIR2" bands as input spectral data for each point

and established a convexity threshold of -6 and 10000 iterations with a standard weighting

exponent of 1, as suggested by Weltje (1997).

We used a pseudo-inverse unmixing algorithm on GEE with two constraints to calculate

fractional covers. The first constraint forces the fractions to sum to one, so that each fraction

represents an actual percentage of each class. The second constraint forces all fractional values to

be non-negative. The resulting maps include three bands corresponding to each of the three

calculated fractions.

2.4.1 Accuracy assessment of fractional cover

We used RCC classifications to assess the performance of our fractional cover estimations, as the

RCC classifications provide very accurate measures of class fractions at the Landsat pixel scale.

Using the results from the classifications performed over VHR imagery, we aggregated the

classified classes into vegetation, BG, and other (including impervious surfaces, water, and cloud

classes). Since NPV is difficult to detect with available VHR datasets, this aggregation permits a separation between vegetation classes (which logically include PV and NPV) and BG, since BG

is the complementary proportion of vegetation when just the two classes occur (i.e., where there is no cloud obstruction, water or impervious surfaces, or: $1 - BG = PV + NPV$). Second, we selected the temporally closest Landsat-based fractional cover layer to a subset of 10 RCC classifications. Third, we generated a layer of the centroids of pixels for these fractional cover estimates and randomly selected 5000 centroids. Fourth, we generated circles of 15m radius (approx. size of

Landsat pixels) at the locations of the sampled centroids and clipped the aggregated classification. From this sample, we only selected the circles fully overlapping vegetation and BG pixels. Fifth, we calculated the proportion of pixels of vegetation and BG within each circle. Finally, after completion of this process, we compared the values of these proportions to the Landsat-derived fractional cover by using regression statistics: $R^2$, normalized root-mean-squared error (nRMSE)

in units of percent cover, and *p*-values.

**3. Analysis**

*3.1 Land cover classification*

Overall, the LCC procedure resulted in an overall accuracy of 85.57 %, with Kappa of 0.832, which is above the recommended threshold of 85 % for LCC predictions and remarkable for such

a large area as our study area (Foody 2002, see Figure 8). The resulting confusion matrix from the accuracy testing partition of the 8,191 randomly selected points is presented in Table 2. The random forest model using all bands was more accurate than those using subsets of input bands. In decreasing order, variable importance derived from the random forest classifier for every band





was elevation, Green, EVI, Red, SWIR2, Blue, Slope, Photosynthetic vegetation, SWIR1,

MSAVI2, horizontal curvature, NIR, and bare ground (Figure 7).

**Table 2**: Confusion matrix of the random forest classifier using multi-year validation samples.

Class codes are presented in Table 1.

| Class | | Reference Data | | | | | | | | Sum | User's Accuracy (%) |
|---|---|---|---|---|---|---|---|---|---|---|---|
| | | CCW | DS | BU | OCW | SS | CL | GR | SV | | |
| Predicted LCC | CCW | 918 | 6 | 0 | 13 | 4 | 3 | 1 | 0 | 945 | 97.1 |
| | DS | 13 | 969 | 15 | 109 | 29 | 6 | 2 | 0 | 1143 | 84.8 |
| | BU | 0 | 26 | 1064 | 138 | 7 | 5 | 3 | 0 | 1243 | 85.6 |
| | OCW | 81 | 18 | 138 | 1080 | 7 | 8 | 4 | 0 | 1336 | 80.8 |
| | SS | 0 | 9 | 110 | 98 | 914 | 14 | 82 | 5 | 1232 | 74.2 |
| | CL | 2 | 3 | 4 | 0 | 4 | 23 | 31 | 0 | 67 | 34.3 |
| | GR | 4 | 3 | 1 | 2 | 109 | 26 | 823 | 9 | 977 | 84.2 |
| | SV | 0 | 0 | 9 | 7 | 3 | 8 | 3 | 1218 | 1248 | 97.6 |
| Sum | | 1018 | 1034 | 1341 | 1447 | 1077 | 93 | 949 | 1232 | 8191 | |
| Producer's Accuracy (%) | | 90.2 | 93.7 | 79.3 | 74.6 | 84.9 | 24.7 | 86.7 | 98.9 | | |

CCW: Closed canopy woodland, DS: Dense scrubland, BU: Bushland, OCW: Open canopy woodland, SS: Sparse scrubland, CL: Cultivated land, GR: Grassland, SV: Sparsely vegetated land.

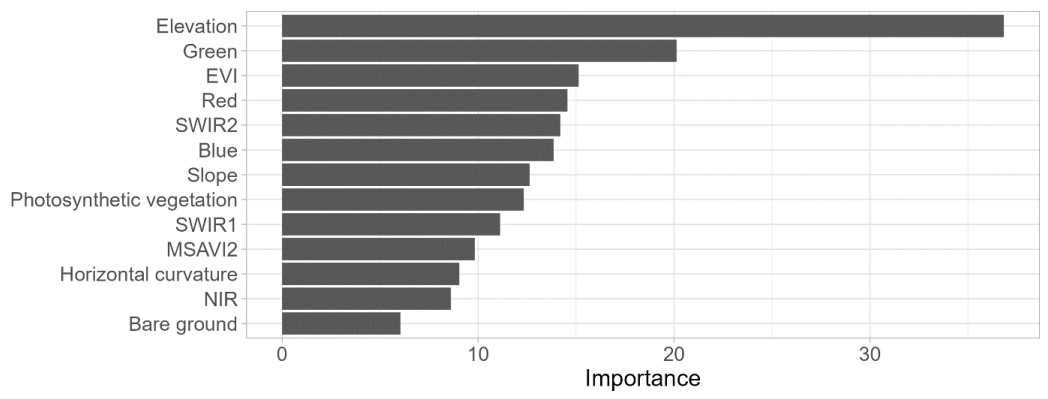




**Figure 7:** Variable importance derived from the best random forest classifier (see description of variables in section 2.3.3).

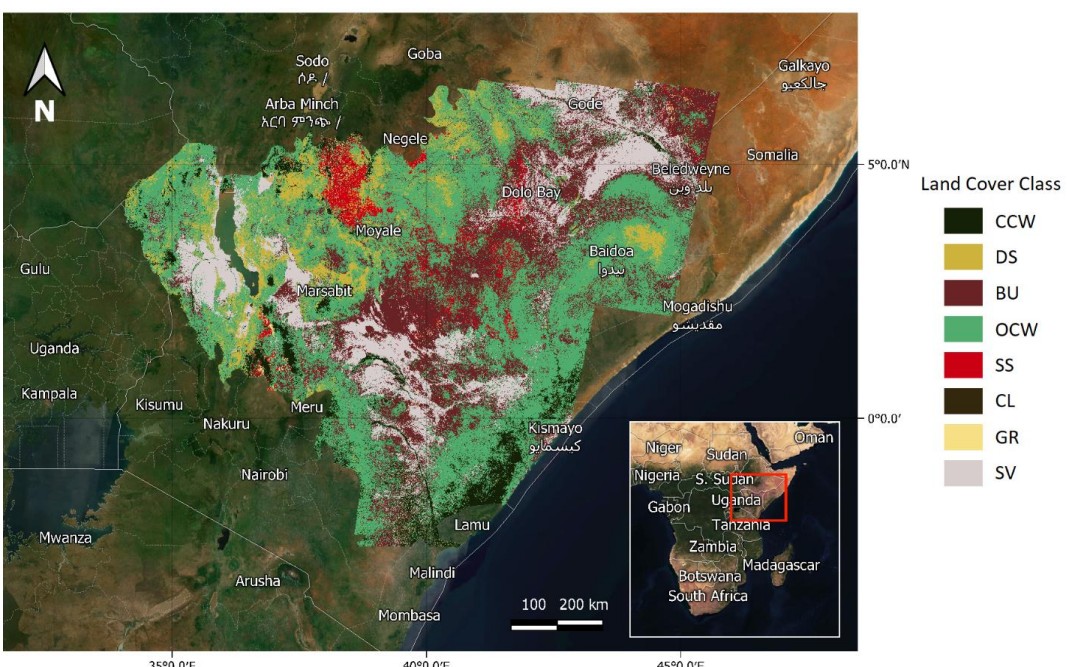

**Figure 8**: 30 m resolution predicted land cover classification for 2015. Class codes and descriptions are presented in Table 1. Basemap: ©MapTiler, https://www.maptiler.com/copyright/.

The annual time-series of the total proportion of each land cover class in our study area, shows

variations in the proportion of SV, SS, and OCW classes around the same years within the studied time frame (Figure 9). To understand the source of such variation, Figure 10 presents the proportion of inter-annual transitions of each pixel from class to class for the study period. Potentially valid transitions are defined in our state transition model, presented in Figure 6. Using this model, we can use the potentially valid inter-annual transitions and compare them with all



inter-annual transitions in each pair of subsequent years (only using unmasked pixels with class

values in both years). Our expected, potentially valid, inter-annual state transitions between land

cover classes (Figure 6), were above 62.30 % in all yearly transitions (Figure 10) with a mean of

75.20 % and a maximum of 83.20 %. The number of unmasked paired pixels as a proportion of

the total Landsat-based pixels used for the calculation of land cover had a minimum of 86.60 %,

with a mean of 95.30 %.

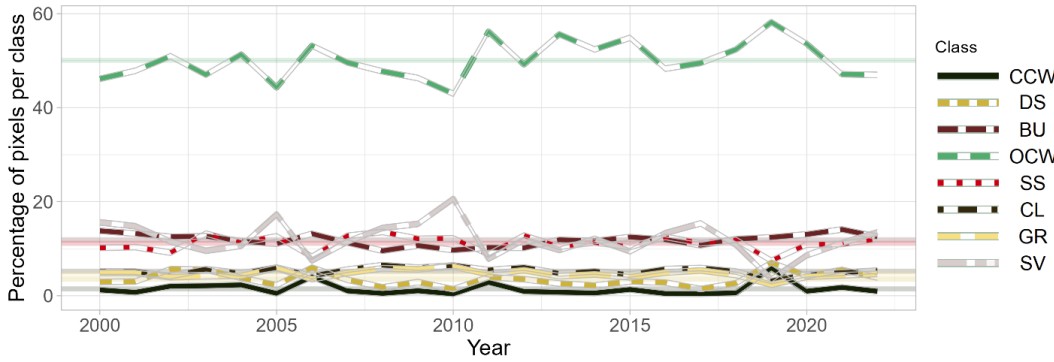

**Figure 9**: Annual time series of proportion of pixels of land cover classes for the entire study area

(pixel count = 858,780,117). Horizontal lines correspond to study area average values for each

class over the study period.

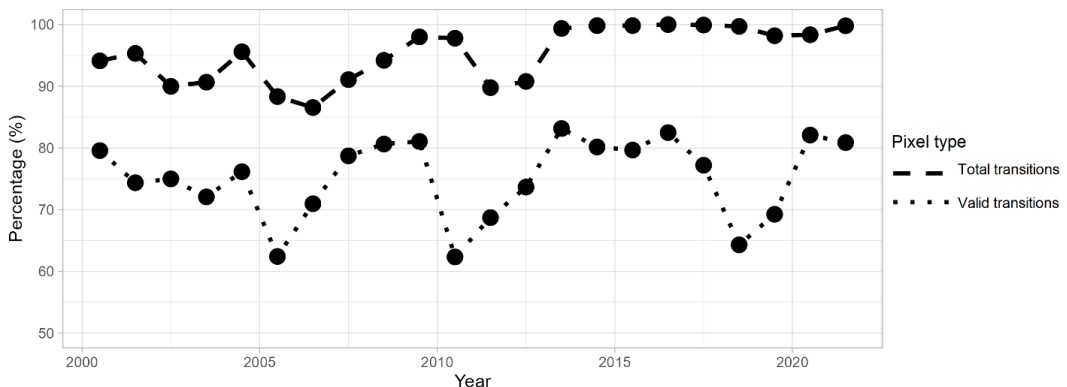

**Figure 10**: Proportion of pixels with potentially valid yearly transitions. Dashed and dotted lines show the total amount of paired unmasked land cover classes, and the total amount of potentially valid transitions as per our state transition model presented in Figure 6.


Filtering out pixels with unlikely transitions as defined in our state transition model, allows to reconstruct the history of individual pixels and help understand their change through time. The alluvial chart is a useful visualization to track such transitions through time by presenting the frequency distributions of classes in different time periods, aggregating the change of pixels with

the same transitions between classes into individual ribbons. Figure 11 shows the decadal change of 48,280 randomly selected pixels with potential valid transitions and no missing data in our study area from 2000-2020. By assigning colors to the last year in the sequence, it is possible to visually track changes, evidenced by the width of the lines moving from one class to another between

periods. The largest change of classes in this sample corresponds to 1.75 % of pixels (n = 845) staying as OCW in 2000 and 2010 but changing to CCW by the year 2020 (see dark ribbon going from OCW to CCW between 2010 to 2020). This is followed by 1.37 % of BU pixels (n = 661) turning into SV by the year 2010 and staying in that class until 2020 (see dark ribbon going from BU to SV between 2000 and 2010). Other classes present changes less than 1 %.


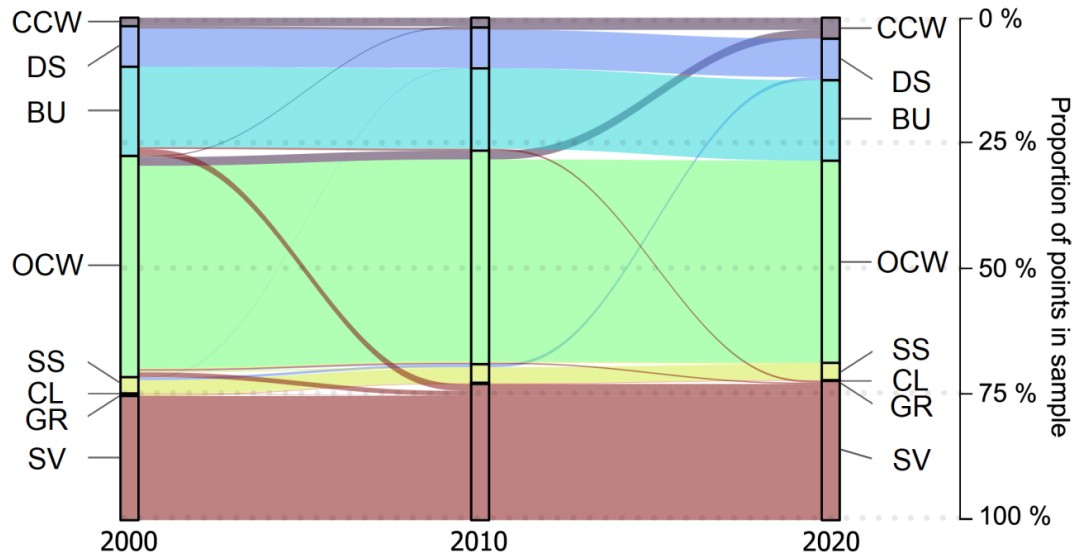

**Figure 11**: Decadal vegetation transition between 2000-2020 of 48,280 random pixels with potentially valid land cover transitions as defined in our state transition model for the three selected years. Land cover classes are presented in Table 1. Color codes were assigned to land cover classes

present in the locations in year 2020 in order to track changes between decades.

*3.2 Vegetation fractional cover estimation*

Endmember estimation reached the threshold convexity error of -6 after 3,265 iterations, with total negative values representing just 0.026 % of the sample, reflecting excellent model fit and a very

small proportion of sample points falling off the multidimensional space between endmembers (Weltje 1997). Figure 12 shows the estimated spectral signatures of endmembers, where a large spike in NIR is visible for PV and high values of reflectance at the SWIR bands are also discernible for BG. Regression results from the comparison between bare ground estimations from HR imagery and Landsat-based predictions yielded $R^2 = 0.795$, $p < 2.2e\text{-}16$, normalized root mean



squared error nRMSE = 0.123, with equation $y = 0.959$ (SE = 0.010) $x + 5.768$ (SE = 0.843), F =

9201.1 on 1 and 2152 DF with p-value: $< 2.2e\text{-}16$ (Figure 13).

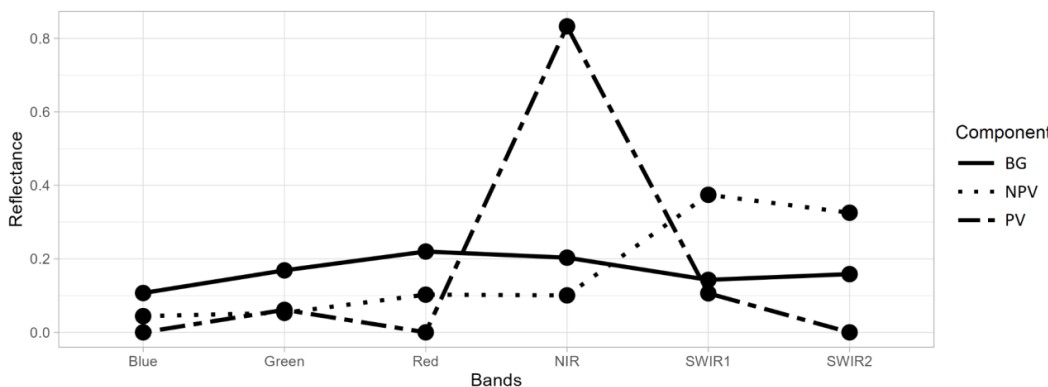

**Figure 12**: Estimated spectral endmembers for fractional cover estimation.


Earth System
Science
Data

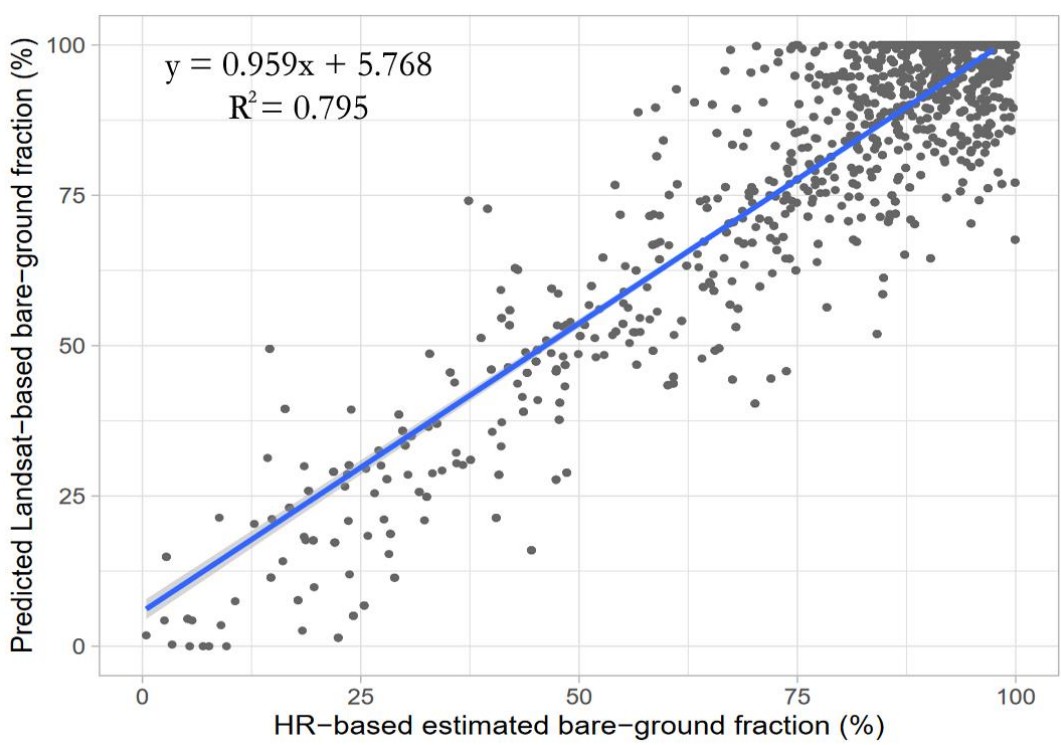

**Figure 13**: Spatial-temporal correlation between HR imagery and Landsat-based predictions of bare ground fractional cover (n = 2,190) at Landsat scale of 30 m from 2016 to 2020. F = 9,201.1 on 1 and 2152 DF with p-value: $< 2.2e\text{-}16$.


Final products consisted of yearly short dry season estimations of fractional cover for our entire AOI with a total of 858,780,117 pixels (Figure 14). Further qualitative assessment of fractional cover predictions against natural color Landsat images and compositions, confirmed accurate representations of the ground conditions. The most quickly identifiable components BG

and PV, show regional accordance with very dry and forested areas, respectively, within our AOI (Figure 14). Similarly, to the LCC time series, fractional cover showed distinct variations in three different periods (Figure 15).



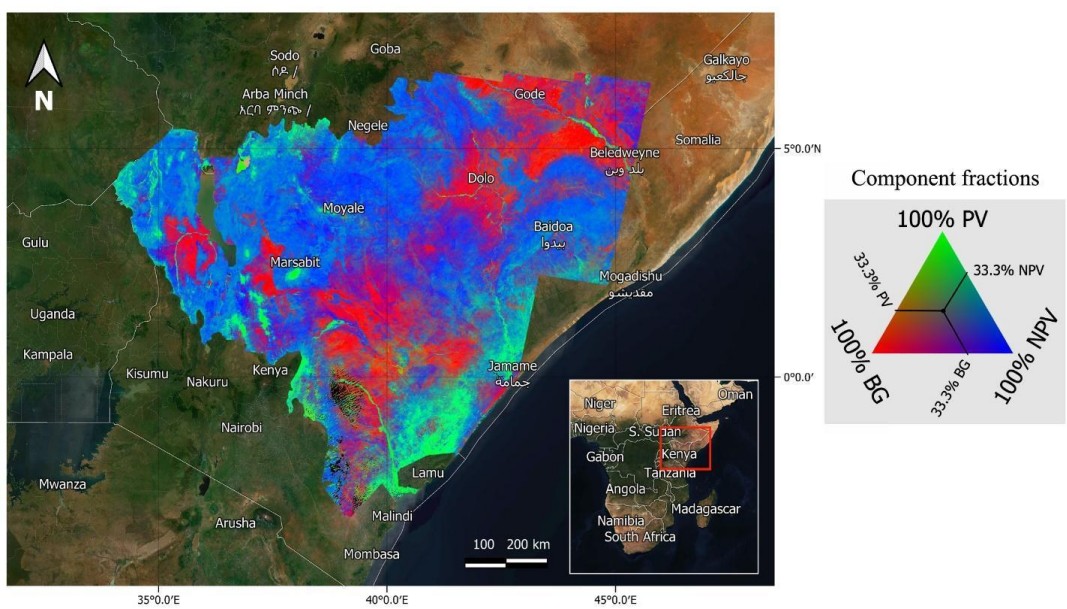

**Figure 14**: Landsat derived 30 m resolution fractional cover estimations for the short dry season of 2020, with mixtures of PV: Photosynthetic vegetation, NPV: non-photosynthetic vegetation, and BG: bare ground for our entire AOI (see legend on figure). Basemap: ©MapTiler, https://www.maptiler.com/copyright/.

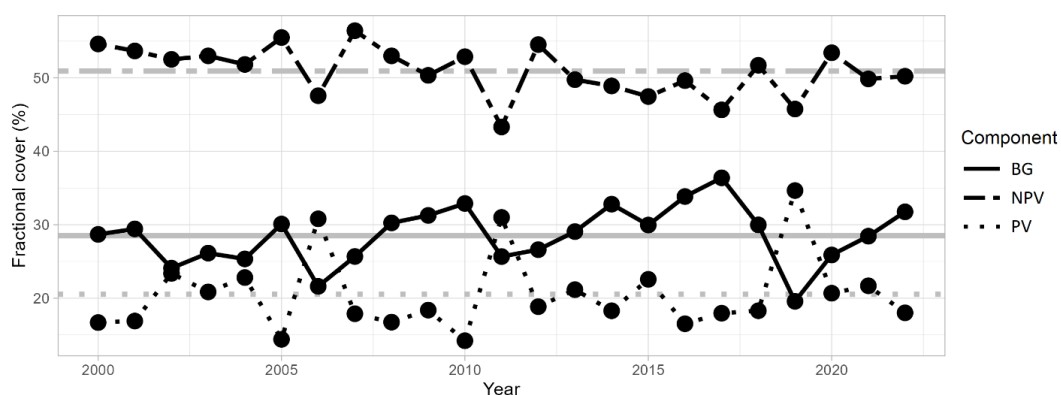





**Figure 15**: Annual time series of average fractional cover values for BG, NPV and PV for the
entire study area (pixel count = 858,780,117). Horizontal lines correspond to study area average
values for each component over the study period.

### 4. Discussion

605        The dataset generated in this study represents a substantial improvement over previously
available data to assess rangeland health in the region, such as plain NDVI from Landsat. These
improvements are the result of a high spatial and temporal resolution, a long temporal extent, and
uses land and fractional cover metrics expressly designed to inform monitoring and assessment of
East African rangeland systems (e.g. Hill and Guerschman 2022, Sexton et al., 2013, Buchhorn et

al., 2021).

           Our land cover classification scheme allowed us to reach acceptable per-class accuracy
levels, using 85 % as a reference value for most of our land cover classes (Mundia and Aniya 2005,
Rogan et al., 2003, Treitz and Rogan 2004, Weng 2002, Yang and Lo 2002), considering the
limitations of both the availability of ground reference data and Landsat imagery. Our proposed

method that used VHR imagery to generate training and validation data for the Landsat-based
classification has proven to be key to reaching these accuracy levels, enabling us to increase the
amplitude of spectral information of the different features found across such a large and
heterogeneous area. VHR imagery also allowed us to have homogeneous spatial representation in
ground-reference data as shown in Figure 2, thus reducing biases from imbalanced sampling

(Carlotto 2009, Elmes et al., 2020). We also included a minimum threshold value for VHR
classifications and applied a ruled-based algorithm to generate training data, therefore helping to
reduce and control our training data error (Elmes et al., 2020, Padial-Iglesias et al., 2021).
Homogenization of the VPI process also helped standardize training data generation, accounting

for the arising inconsistencies that might impact the Landsat LCC estimations (Elmes et al., 2020,

Foody 2009).

One limitation of our product is its comparatively lower classification accuracy for

cultivated land areas. The close spectral correspondence between the dominant cultivated grain

crops in the region (e.g., teff, maize and sorghum in Ethiopia) and wild grasses makes separation

of the two challenging. In addition, other land classes such as sparse shrub could also be difficult

to separate from cultivated land (Hansen et al 2005, Sexton et al 2013), because they are dominated

by either PV or NPV during the short dry season where our Landsat compositions were compiled.

These two factors limit the applicability of the proposed approach to extensive rangeland areas.

We encourage users of this dataset to explore the behavior of the CL class within their study areas

before carrying out further analyses. In addition, cloud cover in this region implies that other tools

such as dynamic time warping (Muller 2007) might not improve land cover estimations, as this

technique requires the extraction of temporal features from time series that are not possible to

generate using Landsat imagery in our defined temporal extent. As with virtually all visible-light

satellite-based remote sensing, cloud cover limits our analysis, both reducing the amount of per-

pixel available imagery, and also the proportion of pixels with available data over our study area.

Other factors such as precipitation resulted in a > 30 % drop in accuracy due to increases in annual

accumulated precipitation, as found in our preliminary classifications.

As shown in Figure 11, this dataset can not only provide descriptions of all the LC pixel

transitions of a given study area but has the potential value of providing a foundation for

assessments of long-term change trajectories that likely will extend beyond the time scope of the

current study. Ecological studies on ecosystem and community dynamics require long-term

ecological datasets (Ellis et al., 2006, Magurran et al., 2010, Ott et al., 2019). Further use of these

products should demonstrate its usefulness as a monitoring, prioritization and inventory tool for planning and decision-making (Allred et al., 2022). Land cover mapping will enable isolating signals from rangelands and incorporate heterogeneity into management frameworks, providing foundations for assessments of long-term change trajectories that likely will extend beyond the time scope of the current study in this specific geographical region (Fuhlendorf et al., 2012).


Vegetation fractional cover estimates showed high accuracy, also attributed to the value of the availability of VHR imagery (Brandt et al., 2020), used for generation of ground reference data for training and validation. Even under our limitations on ground reference data, bare ground, a key indicator of rangeland health conditions for monitoring and management (Pellant et al., 2020, Rigge et al., 2019, 2020), was accurately identified over a relatively large area of more than 4.6 million hectares. Figure 15 shows the potential value of this dataset by presenting a summarization of the annual trend of all three fractional components, which can be reconstructed from different spatiotemporal aggregations, down to the pixel level. Such trajectories will likely help understand the contributing factors for observed and unobserved patterns in the past two decades (Rigge et al., 2021). While further exploration of the spatial and temporal distribution of these trends is needed, this overall assessment might reflect a slow degradation of rangeland condition as bare ground fraction gradually increases (Figure 15).



Here, we used intensive algorithms on VHR satellite imagery to allow training and assessment of the performance of our proposed methods, as little ground reference information exists in this vast and remote region. This helps on maintaining enough detail on the land cover classes and allowed the creation of a relevant VFC estimation. Our maps could help generate new threads of rangeland maps for East Africa, especially to improve community development, ecological conservation, and humanitarian programming. As the lack of ground reference data has


been a bottleneck to empirical rangelands research in this part of the world, our VHR-based

estimations can help develop and improve assessments of rangeland health trajectories. The

increasing availability of remote sensing imagery and the application and development of new

machine learning algorithms will certainly help develop better management tools. Relatively

recent collections such as Sentinel-2 and its harmonization with Landsat imagery (Claverie et al.,

2018) will need to be tested for its advantages and disadvantages for its use in long-term time

series in this geographic area.

Overall, this dataset will be useful to monitor the impacts of different rangeland

management practices or test the impact of development programs. The open access to

sophisticated cloud computing platforms, such as GEE (Gorelick et al., 2017), will contribute to

practical use and further assessment of this dataset. To accomplish this, have made these two

products available in GEE (see Data availability).

## 5. Data availability

Our 30 m resolution annual land cover classification and fractional cover data are publicly

available at https://doi.org/10.5281/zenodo.7106166 (Soto et al., 2023) and Google Earth Engine

(see Appendix A).

## 6. Competing interests

The contact author has declared that none of the authors has any competing interests.

## 7. Acknowledgements

This project is supported by Biodiversity International and the CGIAR Standing Panel on Impact

Assessment (SPIA) for financial support under LoA L20HQ130. YS also acknowledges support

from NASA-CMS award (80NSSC21K1058).



## 8. Author contributions

CB, PC, FF, NK, SW, NJ, CL, BP, and YS conceived the study. PC and CL conducted fieldwork. SW, PC, CL, BP, and GES conducted visual photo-interpretation work. GES performed the remote sensing analyses and wrote the first draft of the manuscript. All authors contributed to discussions and writing of the manuscript.

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



**10. Appendices**

**Appendix A. Description of access to Google Earth Engine (GEE) data.**

**Land Cover Classification** data can be accessed using GEE's asset ids with the following structure:
-   projects/ee-gerardosoto/assets/lcClass<YEAR>
-   For example, for year 2000, use: "projects/ee-gerardosoto/assets/lcClass2000"

Alternatively, use the GEE's links as follows:
https://code.earthengine.google.com/?asset=projects/ee-gerardosoto/assets/lcClass2000

**Vegetation Fractional Cover** data can be accessed using GEE's asset ids with the following structure:
-   projects/ee-gerardosoto/assets/fracCov<YEAR>_int16
-   For example, for year 2000, use: "projects/ee-gerardosoto/assets/fracCov2000_int16"

Alternatively, use the GEE's links as follows:
https://code.earthengine.google.com/?asset=projects/ee-gerardosoto/assets/fracCov2000_int16





**Appendix B. Reference flash card sets.**

The following pages include the flash cards used to reference land cover types and canopy cover.

Figure B1. Flashcard for land cover type "Closed Canopy Woodland".







Figure B2. Flashcard for land cover type "Dense Scrubland".

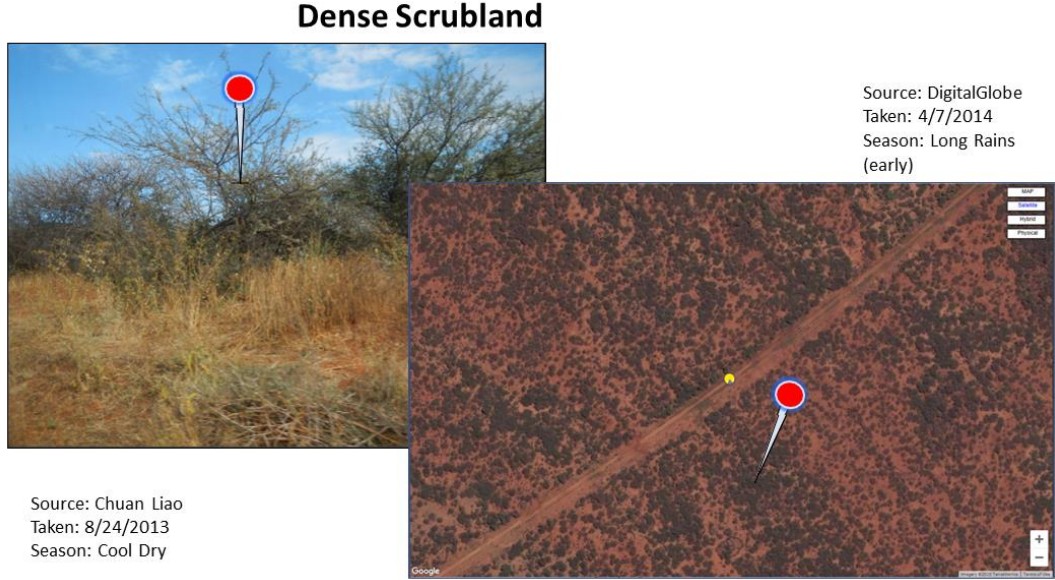




Figure B3. Flashcard for land cover type "Bushland".





Figure B4. Flashcard for land cover type "Open Canopy Woodland".

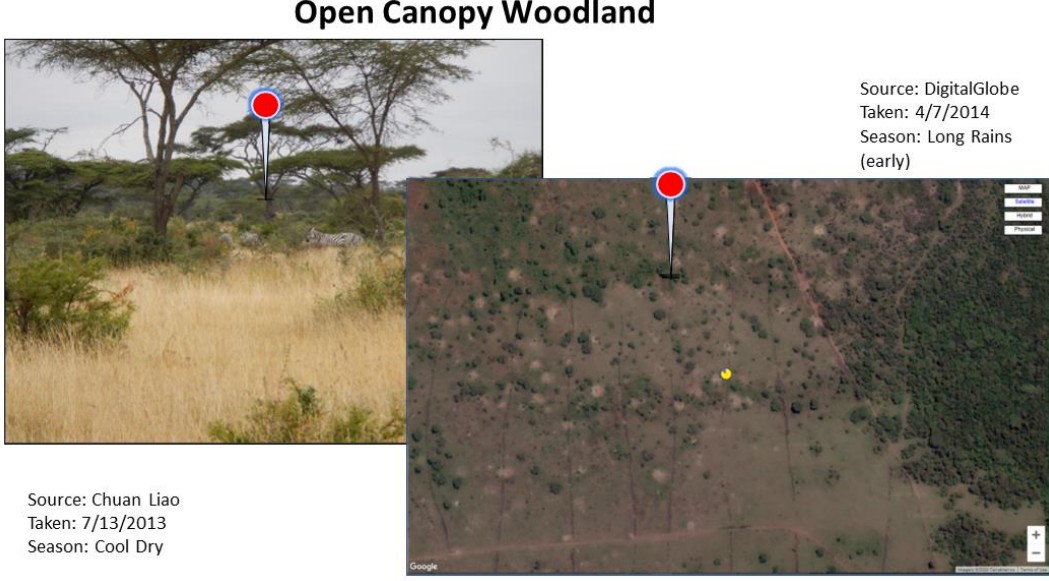




Figure B5. Flashcard for land cover type "Sparse Scrubland".

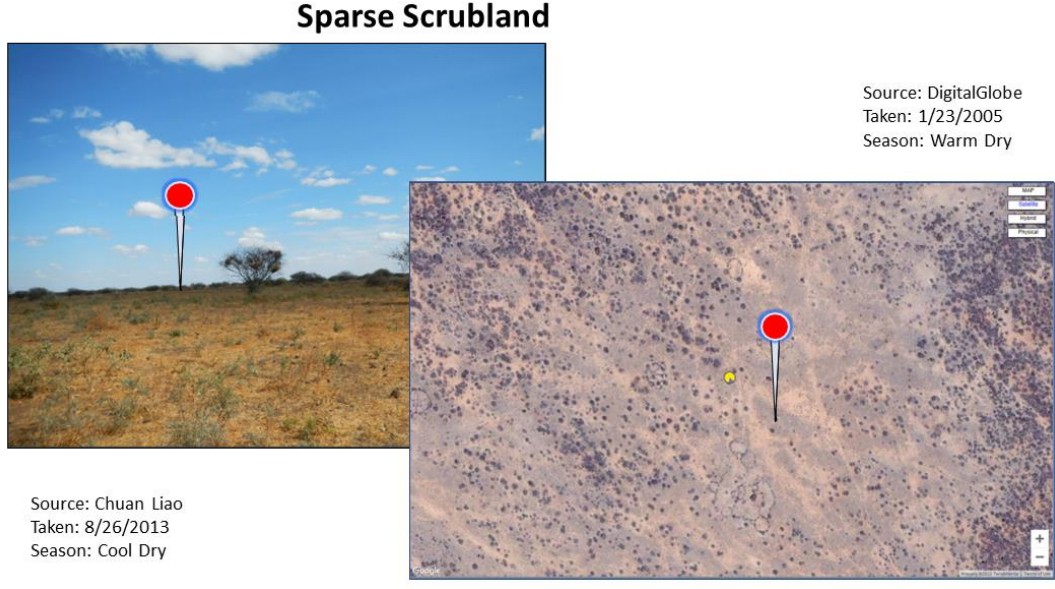



Figure B6. Flashcard for land cover type "Cultivated Land", maize crop.



Figure B7. Flashcard for land cover type "Cultivated Land", cropped versus fallow.





Figure B8. Flashcard for land cover type "Cultivated Land", teff crop.

**Cultivated Land (w/ teff crop)**

Source: DigitalGlobe
Taken: 2/15/2018
Season: Warm Dry

Source: Chuan Liao
Taken: 6/28/2013
Season: Long Rains (late)



Figure B9. Flashcard for land cover type "Grassland".



Figure B10. Flashcard for land cover type "Sparsely Vegetated Land".

**Sparsely Vegetated Land**

Source: DigitalGlobe
Taken: 3/7/2014
Season: Warm Dry

Source: Chuan Liao
Taken: 7/8/2013
Season: Cool Dry (early)






Figure B11. Flashcard for canopy cover level "2m diameter in a 30 by 30 m plot".

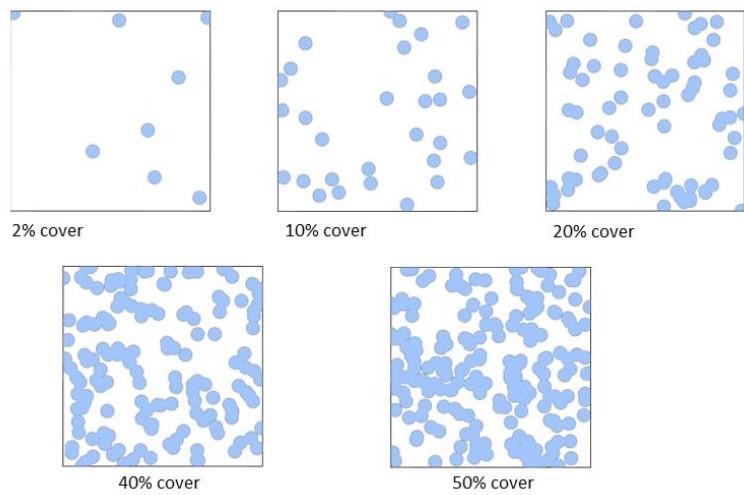



Figure B12. Flashcard for canopy cover level "4m diameter in a 30 by 30 m plot".

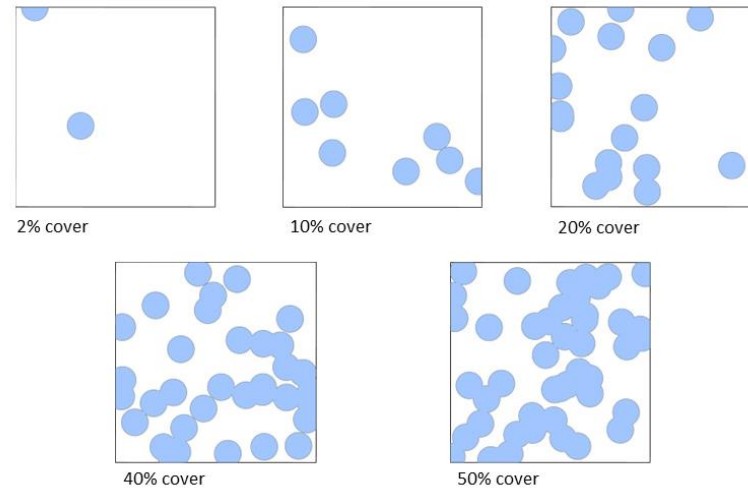





Figure B13. Flashcard for canopy cover level "8m diameter in a 30 by 30 m plot".

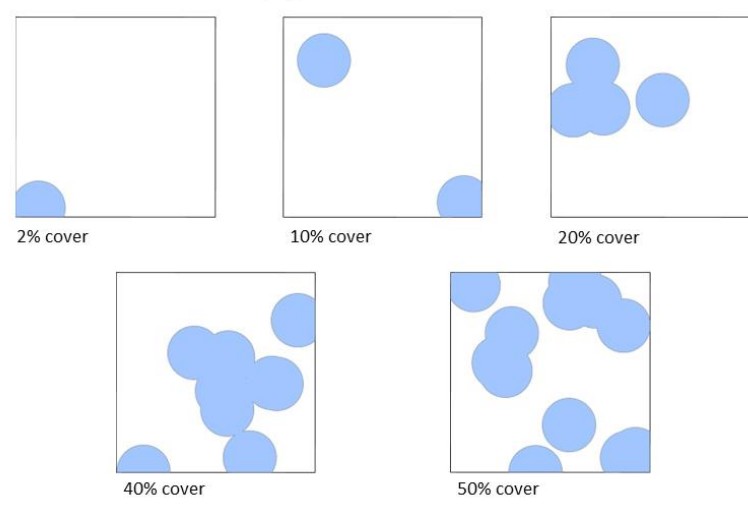






Figure B14. Flashcard for canopy cover level "2m diameter in a 10 by 10 m plot".

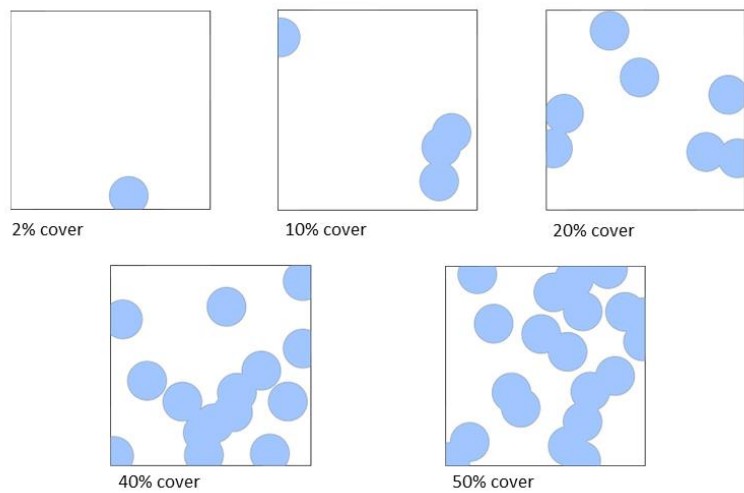




Figure B15. Flashcard for canopy cover level "4m diameter in a 10 by 10 m plot".

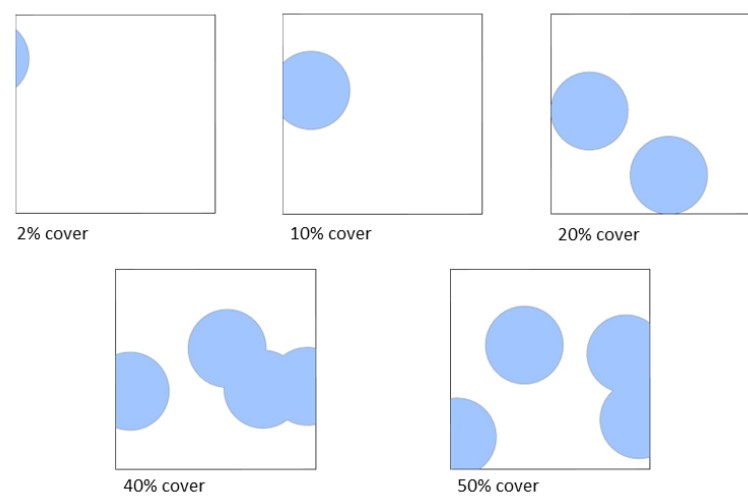




Figure B16. Flashcard for canopy cover level "8m diameter in a 10 by 10 m plot".

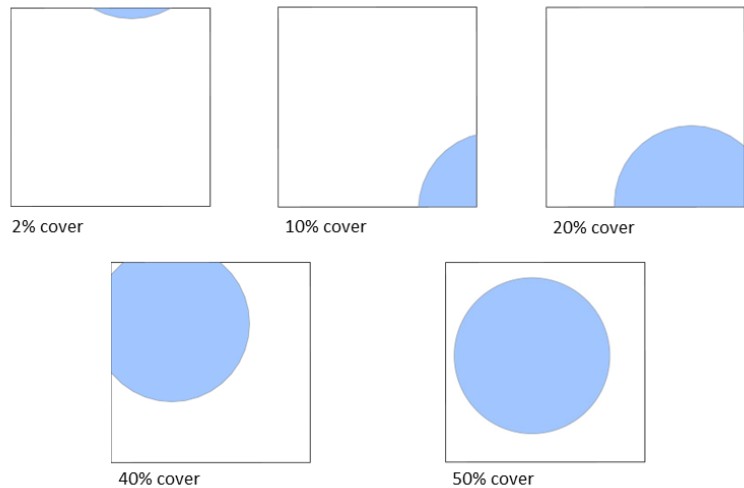