# Peer review of "Mapping Rangeland Health Indicators in East Africa from 2000 to 2022"

_Earth System Science Data, 2023_

## Author Comment (AC1)

**Response letter**
Manuscript with Ref: essd-2023-217 entitled
"Mapping Rangeland Health Indicators in East Africa from 2000 to 2022"

We would like to thank all three reviewers for their constructive and valuable comments. We took all the suggestions into account, which can be found in the track changes version of the revised manuscript. Below we provide a summary of the main changes, followed by detailed explanations of how we addressed each of the reviewers' comments.

Responses to specific comments are presented below.

**Reviewer #1**
This manuscript provides a thorough description of techniques combining very high-resolution imagery, ground photos, and Landsat scale imagery to classify landcover class and fraction cover of photosynthetic veg, non-photosynthetic veg and bare ground in rangeland over a region in East Africa. The methods described here provide a useful template for broader scale implementation.
*Response: Thanks for appreciating the value of this work.*

A few suggestions:

L185: Here it states data over four decades were used in this study. However, the title and Fig. 3 suggest that data from 2000 to 2022 were used. Please clarify.
*Response: Sorry for the confusion. We meant to state that Landsat dataset has been available for four decades, which is an excellent asset for generating long-term time series of land cover types (and other products). However, specific to this study, our products (land cover types and fraction of photosynthetic, non-photosynthetic, and bare grounds) are done for the period of 2000-2022. To address this confusion, we rephrased our statements in the revision as follows: "To capture historical changes in vegetation health in our area of study, we utilized Landsat dataset, which has been available for over four decades (1982-present, Wulder et al., 2012) and thus enables the development of long-term time series of land cover classes and vegetation fractional cover".*

L205-206: The thermal bands are not atmospherically corrected with LaSRC, but rather with MODTRAN. Since the thermal bands are not used in this study, suggest not mentioning them or their correction approach to avoid confusion. Focus discussion on the VSWIR bands.
*Response: We appreciate the comment and removed the references to thermal bands and adapted the text to adjust in context.*

L234: pixels that *were* not included
*Response: Corrected as suggested.*

Fig. 3: Explain the drop in % cloud-cover pixels in 2013.
*Response: Thanks for this suggestion. This sudden drop in cloud cover since 2013 is likely a consequence of the launch of Landsat8 in 2013, which has improved sensor characteristics and data collection capacity. To clarify this, we added a phrase in that section as follows: "The launch of Landsat 8 in 2013 not only implied an improvement in the sensor characteristics, but also increased data collection capacity, thus reducing the likelihood of acquiring cloud-covered imagery as it becomes evident in our study area (Figure 5)."*

L248-294: These are not complete sentences. Suggest: "The methodology to build long-term time series of LCC and VFC for rangelands in Eastern Africa is divided into three major steps: first, the development of a training/testing dataset from VHR imagery; second, the LCC classification; and third, the VFC classification."
*Response: Revised as suggested.*

L254-259: Long sentence. Suggest breaking up. The next sentence is confusing too and could use a rewrite.
*Response: Revised as suggested. The new text reads "To generate the LCC reference data, we generated an algorithm that created reference points using a set of conditions with the proportions of reference compositional component (RCC). The RCCs within each of our LC class definitions include vegetation functional groups and other important classes such as bare ground. The RCCs are then compared to the calculated proportion of pixels from the VHR classification within a moving window matching the 30 m spatial resolution of the HR data."*

L294: Have these 8 classes been tabulated before this point within the main manuscript? If not, suggest doing so or including a table with some primary characteristics. After reading further, I see this in Table 1. Suggest introducing or referencing this table earlier in the text.
*Response: Revised as suggested. We introduced Table 1 earlier in the text (now in section 2.3.1) and cited it when we first mentioned the 8 classes. The new text reads: "Specifically, a team of four VPI analysts* **was trained to identify eight land cover classes following those employed by Liao and Clark (2018). We made additional refinements to these classes as described in Table 1.** *A detailed protocol was developed to ensure effective quality control."*

Fig 10: What specifically is the cause of the low percentage valid transition years of 2006, 2011 and 2018? Was this discussed?
*Response: We added phrases to discuss the causes. Note the previous Fig. 10 is now Fig. 12 in the revision. The new text reads: "Three drops in the number of valid transitions are visible in Figure 12, which correspond to three drought events followed by rains and a greening effect on the landscape (Okal et al., 2020). This effect is consistent with the changing proportions of Closed Canopy Woodland (CCW) and Sparse Vegetation (SV) for 2005-2006, 2010-2011 and 2017-2019 in Figure 11."*

Fig 14: It would be interesting to see a time sequence of this plot, perhaps 2-year intervals? Do annual maps of change in fractional cover highlight regions of interest?
*Response: Thanks for this excellent suggestion. We are currently preparing a separate manuscript that focuses specifically on the trends/variability in a spatially explicit manner. Therefore, we prefer not to include such information in the current manuscript, which is technical data development and description oriented.*

**Reviewer #2**

This study produced a land cover and vegetation fractional cover data in the arid and semi-arid Kenya, Ethiopia, and Somalia using a machine learning approach. The logic of the manuscript is clear but there are a few points that need to be cleared. First, this is a typical land cover classification study and the ecological implications are limited at this stage Thus, the "health indicators" in the title seem confusing. Where did the manuscript describe the implications of the ecological aspects of rangeland health? From what I have read so far, it provides a spatial-temporal distribution of different land covers. If rangeland health needs to be included, readers would like to see the driving forces analyses. Second, the methodology part is the most important part but it is not clear. In section 2.1, could you explain more to clearly show the three steps? Specifically, how were the reference data generated and used, given the importance and the intensive labor needed to compile such a dataset? I am curious also because I am not sure how these samples could be used to train the model to separate some of the very similar land cover types.

*Response: We appreciate all the critical comments raised by this reviewer.*
*First, regarding the "ecological implications are limited" and the suggested "driving forces analyses" (which are excellent suggestions), we are actually working on separate manuscripts focusing exactly on these issues. The two products developed here are the key "building blocks" for assessing rangeland health from the ecological and economic perspective, as described in our Introduction. We intended to separate the technical aspect of data product development and science analyses/interpretation into different papers; otherwise, the scope of the paper would be formidable.*
*As such, we present these datasets as tools for assessing rangeland health trajectories, as a data-oriented paper, which also aligns with the aims and scope of ESSD. In the meantime, we have a couple of manuscripts in the pipeline that carried out detailed analysis on the possible causes of the patterns described here.*

*Second, regarding the clarity of the methodology, we have included the overall strategy of our technical workflow at the beginning of the method section to help logical flow and clarity of this manuscript.*

*Third, regarding the creation of reference data, we proposed a very detailed and rigorous procedure to create reference data that can separate classes based on compositional components. We believe the restructuring of the method section will help readers better understand our three-step approach to create reference data and produce the two relevant data products. We also have included many improvements to the previous version thanks to this reviewer, which we believe will clarify the doubts about separability of classes and spatial distribution of the reference dataset.*

Other minor suggestions:

Line 164, the full term of NGA should be shown when the abbreviation appears for the first time.
*Response: Full name included as suggested.*

Line 165: HR or VHR?
*Response: Thanks! Corrected to VHR.*

Line 175, what is SD?
*Response: Clarified to "short dry".*

Lines 175-178: Model trained by the samples during this period has been applied for classifying historical land cover characteristics. This approach was called transfer learning in some studies, which reported that it might have potential problems in capturing land use changes. Will the study area also be affected?

*Response: Thanks for this great point. We are fully aware of this issue and should have included some discussion. While the term "transfer learning" is often used in deep learning applications, in the revision, we included a dedicated section to discuss these possible effects:*

*"In addition to class-specific issues, the multi-year classification scheme used here, has limitations and possible effects on the classification accuracy for years when reference data were not available. This study does not explore this effect due to the lack of in-situ reference data for the total length of the studied period. However, future studies in similar ecosystems where reference data is available are needed to further improve classification performance. Current research on the use of transfer learning, with the use of pre-trained models and fine-tuning with limited local data provides promising opportunities for further improvement of remote sensing products and possible bias exploration (e.g. Li et al. 2023 , Račič et al. 2024, Weikmann et al. 2021).".*

Line 179, "area of interest": AOI
*Response: Corrected to "area of interest".*

In section 2.2, could you show the spatial distribution of the samples, including the groups used for model calibration and validation, as well as the LC types?

*Response: We really appreciate this comment as it made us realize we did not reference the spatial grid used to classify RCC data anywhere in the text. The spatial distribution of reference samples follows our point grid presented in Figure 3. Thus, we incorporated a reference to this figure in section 2.3.1 as follows: "This additional VPI work followed a procedure to spatially label the key components within each of the land cover classes and was focused on a grid of 8 x 8 km squares centered in a regular point pattern where VHR imagery was available (see Figure 3).". In addition, we created a figure showing the proportion of reference data grouped by land cover classes and elevation intervals, Fig. 10. In addition, we added a phrase at the end of section 2.4.3 to provide the percentage of the validation random sample relative to the total sample of reference data. The phrase reads: "The percentage of classes in the random validation sample relative to the total amount of reference data was 9.3% for CCW, 14.5% for DS, 13.6% for BU, 7.7% for OCW, 9.6% for SS, 1.7% for CL, 8.8% for GR, and 16.7% for SV (see Table 1 for class names).".*

Figure 9, suggest to use color and line type to show the trends. Also, the trends could be added.
*Response: Suggestion adopted. We have included colored lines and added linear trends for each land cover class within the revised figure, now Fig. 11.*

Figure 15, show trends please
*Response: Suggestion adopted, see our response above.*

Line 607, this is an annual product. What do you mean by the high temporal resolution here? Which dataset did you compare to?

*Response: Sorry for the confusion. In the revision, we removed "temporal" from the statement. Now it reads: "These improvements are the result of a high spatial resolution, a long temporal extent, and use of land and fractional cover metrics expressly designed to inform monitoring and assessment of East African rangeland systems". Also, we provide a list of references for the second question, but we also included MODIS products in the first sentence, which now reads: "The dataset generated in this study*

*represents a substantial improvement over previously available data to assess rangeland health in the region, such as plain NDVI from Landsat **and MODIS products**.*

**Reviewer #3**

The authors present a remote-sensing derived dataset on rangeland health for a region in Africa. While I cannot judge the merit of the published data, the methods are well described, and the methodology seems sound.

It remains somewhat unclear why the specific study area was chosen, and not a larger area (e.g., all of Sub-saharan Africa). A larger area would increase the usefulness of the published data.

*Response: Thanks for appreciating the value of this work. Our rationale for choosing the domain of this specific part of East Africa were first and foremost that this was a part of a broader effort aimed at evaluating the rangelands impacts of a development intervention within this space in northern Kenya and southern Ethiopia. The project funding for that effort permitted covering an unprecedentedly large area of African rangelands, but not the whole of East Africa, much less the whole of Sub-Saharan Africa. That said, this area likely has some broad generalizability in that these rangelands: 1) represent a vast rangeland/dryland ecosystems that well represent hyper complex and rapid physiological/phenological dynamics in other regions of the world (Adams et al., 2021; L. Wang et al., 2022), 2) it remains a data "desert" of ground data for model training/validation, and 3) this region suffer strongly from climate change and extremes (e.g., droughts, floods, etc, IPCC, 2022) and their consequences on rangeland health, resilience, and well-being of pastoralists (Pricope et al. 2013; Beal et al. 2023). Therefore, East Africa serves as an ideal testbed to tackle all three challenges, which may also plague many other regions of the world. Fortunately, our team collected a sizable amount of local data in this data desert region, facilitating the development, training, and validation of our framework. In the revision, we have A) clarified this rationale in the study area section and B) added the implication of the generalizability of our framework and future research directions in the discussion.*

I recommend a minor revision.

Further comments:

Fig. 2: very nice visualization of the spatio-temporal distribution of VHR data scenes used for training data generation. Would it be possible to provide a similar figure for the Landsat scenes used for inference?

*Response: We have added Fig. 3 to show the spatial distribution of Landsat tiles used for the analysis. We also describe the use of a precipitation mask applied to these scenes to reduce classification error in section 2.1, which reads as follows "We used two main features to bound our study area. To the east and north, we used Landsat tiles, using PATH 164 and ROW 56 as limits, dropping tiles PATH 164, ROW 59 and 60 due to heavy cloud cover. To the west and south, we used a threshold value of mean annual precipitation of 700 mm using TerraClimate data (smoothed with a kernel convolution with Standard Deviation = 5 km; Abatzoglou et al., 2018), thus keeping the focus on the rangeland-dominated arid and semi-arid areas.".*

Fig. 7: It is surprising that elevation has the highest feature importance. Is there a strong dependence between class occurrence and elevation? This could be tested in a quick analysis, based on the elevation at the training / validation samples? (e.g. showing stacked barcharts of the class proportions per elevation slice).

*Response: This is a great point. There is actually a correlation between elevation and the most represented classes in the study area; for example, lowlands are mostly drylands in this region. In fact, closed canopy woodlands (CCW) are described in our classification as upland vegetation types.*

*Following the reviewer's suggestion, we added a new figure showing the proportions of reference data in 400m elevation bands for each of the eight classes.*

Also, what are the implications of this regarding model generalizability?
*Response: This is an excellent point. We expect our models specifically trained in our study domain will be generalizable to other dryland/rangeland regions in the whole of Sub-Saharan Africa or other continents (e.g., Australia, parts of Central Asia) where similar ecosystems, land cover classes, and herding intensities, etc., exist. At this point, we cannot confirm or refute this generalizability without rigorous validation with local data outside of our study domain, which calls for future studies. However, we are confident that our proposed 3-step framework (esp. harnessing VHR images to generate training labels) can be highly generalizable and easily adapted to other regions of the world and to other domains of science (e.g., crop type classification in the global south). In the meantime, the significant large language model (LLM) or foundation models carry the huge promise to improve generalizability to improve classification accuracy in the complex landscapes. We have added a dedicated section to discuss potential future research directions.*

Would your model also work in flat areas? It would be nice to read some more elaboration on generalizability of this method for other study areas - any expansion planned for the whole of sub-saharan Africa, or similar?
*Response: Regarding the generalizability, please check our response above. Regarding the topography, our model should work in any topographic landscapes (i.e., generalizable), as we did not find any correlation between elevation or slope and misclassified pixels. Misclassifications occurred mostly because of the temporal availability of imagery, crop rotation timing and changes in precipitation, as reflected in the discussion section. Also, in our study area, there is extensive flat area (e.g., Ethiopian plateau). Actually, the slope variable was explicitly incorporated into our model and scored mid in the importance list.*

Table 1, Fig. 8: Could you spell out the class names (and maybe rotate them in the confusion matrix)? It is hard to read this table / Figure when the reader has to switch between tables to decypher the class names.
*Response: Unfortunately, there is not enough space to include full names within Table 1 and all figures; but we included the acronyms in the figure and table captions where possible to improve clarity.*

Fig. 14. Great visualization with striking spatial patterns.
*Response: We appreciate the comment.*

---

## Author Response (AR2)

**Response letter**

Manuscript with Ref: essd-2023-217 entitled

"Mapping Rangeland Health Indicators in East Africa from 2000 to 2022"

We would like to thank the editor for the constructive and valuable comments. We took all the suggestions into account, which can be found in the track changes version of the revised manuscript. Below we provide detailed explanations of how we addressed each of the reviewers' comments.

L85: "Recent scientific advances create an opportunity to map rangelands health using satellite imagery to monitor changes rangeland health" – there's some repetition in this sentence, suggest revising.

Response: Thanks for the suggestion. The text now reads "Recent scientific advances create an opportunity to map rangeland health using satellite imagery to monitor changes at ecologically meaningful scales for landscape planning and management (Allred et al., 2022)."

L125-126: Typically Landsat is referred to as "medium resolution", to distinguish from "high resolution" commercial imagery. "Very high resolution" is more applicable to UAV data.

Response: We agreed with this comment and updated the "HR" references to "Landsat-based" or "Landsat-resolution" depending on the context of the phrase in the manuscript. Accordingly, we changed "VHR" to "HR" throughout the document. We also found other inconsistencies such as the "HR" references in the accuracy estimation for fractional cover, which should have been VHR. They were kept to HR as this is the correct reference for the current version.

L151: include

Response: Corrected as suggested.

Fig 3 caption: Should this be "very high resolution" rather than high resolution?

Response: Following the changes made and reported in the previous response, we kept this caption.

L230: Utilized *the* Landsat dataset

Response: Corrected as suggested.

L238: it *is* available

Response: Corrected as suggested.

L238 and L243: Be consistent … Landsat data is or Landsat data are… Since "data" is plural, I prefer "are".

Response: Corrected as suggested.

L252: uses data *comprised*

Response: Corrected as suggested.

L259-266: This is a repetition of the previous paragraph. Revise.

Response: We disagree with this claim, as Landsat 5 and 7 are treated differently to match Landsat 8 data in the relatively new Collection 2. The only part that matches for both to make interoperability and harmonization easier is the use of the CFMask algorithm to generate the QA_PIXEL and QA_RADSAT bands. We kept this part for both descriptions to be consistent on how we presented the datasets and because we mentioned one of this bands later in the text, which is commonly used as a shortcut to mask for clouds.

L270: tile)*;* otherwise, data

Response: Corrected as suggested.

L328: The software package Nikon… (remove comma)

Response: Corrected as suggested.

L682: scheme used here has… (remove comma)

Response: Corrected as suggested.

The schematic workflow should be redesigned. The current figure is not aesthetically pleasing. Additionally, there are inconsistencies in the symbols used (some are large, while others are small, and they vary in shape even within the same category), and some of the categories are misleading. For instance, why are there separate categories labeled 'input data' and 'RS data'? Isn't 'RS data' a subset of input data? Please revise the chart to depict a more logical and comprehensible flow, such as processes arranged from left to right or top to bottom.

Response: We redesigned the workflow figure and tried our best to make it more readable and aesthetically pleasing. We removed some redundant nodes and used three frames to group the important parts of the workflow. We hope this version meets the expectations.